# Structural Entropy Guided Agent for Detecting and Repairing Knowledge Deficiencies in LLMs

**Yifan Wei[1,2], Xiaoyan Yu[3], Tengfei Pan[2], Angsheng Li[1][†], Li Du[2][†]**

[1]State Key Laboratory of CCSE,School of Computer Science and Engineering,Beihang University
[2]Beijing Academy of Artificial Intelligence, [3]Beijing Institute of Technology
weiyifan@buaa.edu.cn, angsheng@buaa.edu.cn, duli@baai.ac.cn

## Abstract

Large language models (LLMs) have achieved unprecedented performance by leveraging vast pretraining corpora, yet their performance remains suboptimal in knowledge-intensive domains such as medicine and scientific research, where high factual precision is required. While synthetic data provides a promising avenue for augmenting domain knowledge, existing methods frequently generate redundant samples that do not align with the model's true knowledge gaps. To overcome this limitation, we propose a novel **S**tructural **En**tropy-guided Knowledge Navig**ator** (SENATOR) framework that addresses the intrinsic knowledge deficiencies of LLMs. Our approach employs the Structure Entropy (SE) metric to quantify uncertainty along knowledge graph paths and leverages Monte Carlo Tree Search (MCTS) to selectively explore regions where the model lacks domain-specific knowledge. Guided by these insights, the framework generates targeted synthetic data for supervised fine-tuning, enabling continuous self-improvement. Experimental results on LLaMA-3 and Qwen2 across multiple domain-specific benchmarks show that SENATOR effectively detects and repairs knowledge deficiencies, achieving notable performance improvements. The code and data for our methods and experiments are available at https://github.com/weiyifan1023/senator.

## 1 Introduction

With the pretraining process on massive-scale corpora, Large Language Models (LLMs) capture abundant knowledge and demonstrate impressive performance on various downstream tasks (Chen et al., 2015; Liu et al., 2021). However, their performance may still be unsatisfactory in certain knowledge-intensive domains such as medicine and scientific research. This is primarily due to the difficulty in acquiring and scaling up high-quality domain-specific corpora (Lu et al., 2024; Wang et al., 2024), which hinders the ability of the models to handle tasks that require high factual precision.

The development of data synthesis technology (Wang et al., 2023; Zhao et al., 2024) offers an alternative way to address these limitations in remedying the knowledge deficiency of LLMs. While promising, the efficiency of data synthesis remains a significant challenge. This is because current data synthesis methods may not consider the model's knowledge boundaries (Jiang et al., 2021; Mallen et al., 2023; Yue et al., 2025), resulting in substantial efforts spent in generating data that the model may already be familiar with. In fact, even with advanced prompt engineering (Wei et al., 2022; Liu et al., 2025), generated outputs tend to skew toward high-frequency distributions seen in pretraining data, leading to severe redundancy. Therefore, efficient data synthesis should be tightly coupled with mechanisms for effectively detecting knowledge deficiencies (Xiong et al., 2024; Song et al., 2025) within LLMs, so that the synthesized data can repair the knowledge deficiencies.

---

[†]Corresponding Authors.

However, the knowledge boundaries of large models can be quite complex. Although these models are trained on massive amounts of data, their knowledge is implicitly encoded in model parameters (Geva et al., 2021; Wei et al., 2025) rather than being explicitly stored, leading to unclear distinctions between known and unknown information. In specialized domains, this challenge is compounded by the generation of unreliable or contradictory content (Yang et al., 2024c), which produces flawed synthetic samples that hinder the effective expansion of high-quality, domain-specific corpora.

To overcome the aforementioned challenges, we propose SENATOR, a **S**tructural **En**tropy-guided Knowledge Navig**ator** framework, which achieves knowledge deficiency remediation through a closed loop of structured knowledge probing and targeted synthetic data generation. The framework comprises two key components: 1) Knowledge Deficiency Detection: Human-annotated knowledge graph (KG) systematically describes the underlying complexities and intricacies of the domain. However, the combinatorial explosion of possible paths makes enumeration computationally infeasible. To efficiently detect the knowledge paths, we drive the LLM as an agent to explore upon the KG in a Monte Carlo Tree Search (MCTS) manner (Metropolis and Ulam, 1949), with the structure entropy as reward. The Structure Entropy (SE) (Li and Pan, 2016; Li, 2024) metric quantifies the structural information contained within a graph by capturing its topological organization and the interactions among nodes. This provides insight into the model's uncertainty along knowledge paths in the KG. By employing MCTS within the knowledge space, our framework uses SE values as intrinsic rewards to decide whether to expand specific entity nodes, effectively prioritizing the exploration of paths with high uncertainty and detecting critical knowledge deficiencies. 2) Knowledge Synthesis and Repair: Leveraging the critical knowledge paths identified via MCTS, our framework generates synthetic data by employing prompt templates to structure the content. The KG serves as a trusted source to ensure both the data inputs and the synthesized outputs are credible and contextually relevant. This synthetic data is then used to fine-tune the model through supervised learning, enabling continuous self-improvement and effective remediation of knowledge deficiencies.

Our experiments demonstrate that the SENATOR framework effectively detects knowledge deficiencies in large language models and efficiently repairs them, leading to significant performance improvements across multiple domain-specific benchmarks. Data distribution analyses confirm that our synthetic data incorporates knowledge deficiencies from the pretraining corpus. Moreover, supervised fine-tuning (SFT) of LLMs like Llama-3 (Grattafiori et al., 2024) and Qwen2 (Yang et al., 2024b) using this data led to significant performance improvements, demonstrating that targeted injection of missing knowledge can substantially enhance overall model performance.

## 2   Related Work

**Knowledge Deficiency Detection of LLMs**   Though LLMs possess extensive knowledge, they often struggle to accurately delineate what they know from what they do not (Yin et al., 2023; Ren et al., 2023). Several approaches (Jiang et al., 2020; Mallen et al., 2023; Wei et al., 2024)construct knowledge probability distributions based on existing annotated data, using metrics such as answer correctness or confidence scores to assess a model's knowledge proficiency. One line of work (Wei et al., 2022; Li et al., 2023a; Tian et al., 2024a) directly toward enhancing a model's ability to fully leverage its existing knowledge, thereby reducing the proportion of "Unknown Knows". Another line of work pay attention to enabling models to explicitly acknowledge their knowledge gaps, thus minimizing the occurrence of "Unknown Unknowns". Approaches such as R-tuning (Zhang et al., 2023) utilize labeled data with supervised fine-tuning to judge response correctness, while reinforcement learning based strategies have also been explored (Yang et al., 2023b; Kang et al., 2024). In contrast, our approach for deficiency detection is designed not to rely on pre-existing labeled data, but instead to actively explore the KG to detect intrinsic model uncertainty.

**Model Self-Improvement**   Self-improvement methods of LLM focus on leveraging internal knowledge and feedback to iteratively enhance the performance of LLMs (Zelikman et al., 2022, 2024). A pivotal challenge is generating a reliable critique signal to discern high-quality responses from suboptimal ones. Previous methods (Bai et al., 2022; Wang et al., 2023) involve prompting the LLM to generate diverse task-specific queries and corresponding outputs, followed by the application of manually crafted heuristic rules, such as filtering based on query length to remove redundant or low-quality data pairs. Given the complexity of devising effective heuristics, subsequent research (Sun et al., 2023; Li et al., 2023b; Guo et al., 2024) proposes a few general principles or judging

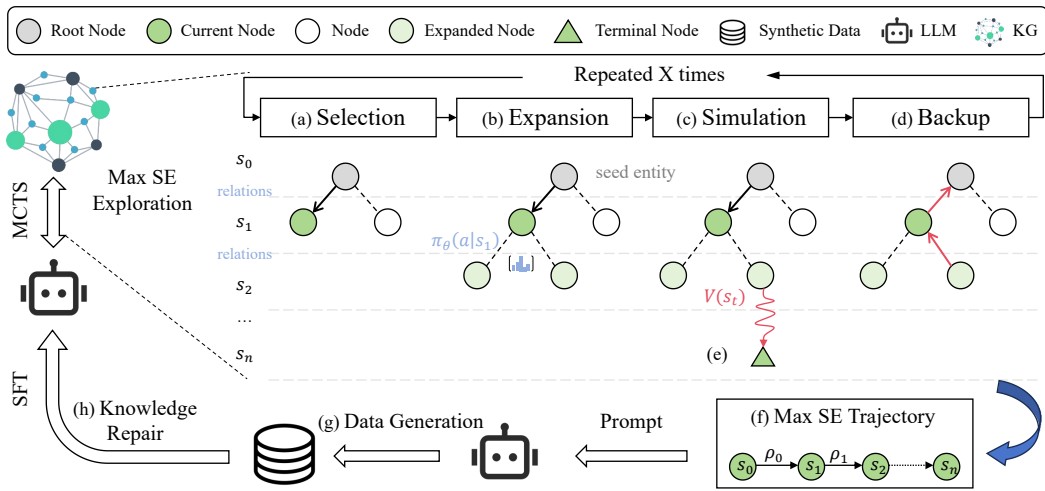

Figure 1: The SENATOR framework operates as follows: An entity state in the knowledge graph is (a) selected, (b) expanded, and (c) simulated using the LLM agent until a terminal node is reached. Specifically, we employ a random policy $\pi$ during the expansion phase. (d) Subsequently, signals from the value function $V(\cdot)$ are backpropagated. This process is iterated multiple times, with the MCTS algorithm searching for (f) better trajectories guided by (e) signals from structural entropy to (g) generate data addressing knowledge deficiencies, (h) and repair model knowledge.

criteria and ask the LLM itself to assess the quality its responses according to these guidelines. However, this approach demands that LLMs possess a robust capability to apply these principles to each specific instance and render accurate judgments. Recently, reinforcement learning-based model show impressive reasoning ability by learning the experiences obtained from explorations in the solution space (Tian et al., 2024b; Goldie et al., 2025). While the probability of obtained plausible solution space of knowledge intensive tasks would be rather limited as the LLM may not possess the necessary knowledge, which would severely restrict the efficiency of exploration and data generation. In this paper, we choose to guide the exploration process in knowledge space using KGs, in a MCTS manner, so as to enable targeted synthetic data generation for high efficiency LLM self-improvement.

## 3 Methodology

Given a knowledge graph, the number of possible knowledge paths $\mathcal{P}$ (i.e., Figure 1f) increases in a combinatorial speed along with the size of KG, making enumerating all possible paths and detecting the uncertainty of LLM on these paths computationally infeasible. To tackle this challenge, as shown in Figure 1, SENATOR employs MCTS to navigate the LLM-based agent to search on the KG for seeking out the most informative paths. To steer the agent to search toward regions with high uncertainty, we introduce a structural entropy based reward function. Based on the identified high-uncertainty paths, data are synthesized to remediate the identified knowledge deficiencies.

### 3.1 Structural Entropy Guided Knowledge Deficiency Detection

The structural entropy based reward function combines the uncertainty of LLM on individual KG triplets with the topological structure information of the KG, guiding the LLM-based agent to perform MCTS over the KG and discover knowledge paths with critical deficiencies.

**Self-Information for Measuring Triplet-Level Uncertainty**    Self-Information (Shannon, 1948) quantifies the amount of information conveyed by a "fact" given its probability distribution. In KGs, a "fact" is represented as a triplet $\tau = <$ subject $u$, relation $\rho$, object $v >$. To measure the LLM's uncertainty of such "facts", we transform $\tau$ into a cloze statement form. The cloze context is formed by combining the subject $u$ and the relation $\rho$, creating a prompt to predict the missing object $v$. The

self-information of a fact $\tau$ is defined as:

$$I(u, \rho, v) = -\log_2 P(v \mid u, \rho), \tag{1}$$

where $P(v \mid u, \rho)$ is the probability of the output $v$ conditioned on the cloze context. Since the relation $\rho$ in KGs is directional, the self-information calculated in this manner serves as a measure of the factual knowledge confidence for the entire triplet.

**Structural Entropy of Modeling Knowledge Path-Level Uncertainty**    To integrate the uncertainty of all triplets along a knowledge path while considering their structural importance, we adopt structural entropy (SE) as a more comprehensive measure of an LLM's knowledge confidence, as shown in Figure 1e. Structural importance reflects the topological significance of a triplet $\tau$ within the knowledge graph. Triplets involving highly connected entities are considered more central, as these entities participate in more relational paths and exert broader influence across the graph. Unlike self-information or Shannon entropy, structural entropy accounts for the knowledge graph's topological structure and the interdependencies among its elements. This is crucial because each triplet is not an isolated piece of information but part of a structured network. The relationships among entities contribute to the overall representation of knowledge. Given a knowledge graph $G = (V, E)$, each edge $\rho \in E$ is assigned a weight derived from the self-confidence in Equation 1. The weighted degree of an entity node $u \in V$ is defined as:

$$d_u = \sum_{v \in \mathcal{N}(u)} I(u, \rho, v), \tag{2}$$

where $\mathcal{N}(u)$ denotes the set of neighbors of entity $u$ and $d_u$ represents the overall uncertainty contained within the node. To quantify the average information content of the graph $G$, we define the one-dimensional structural entropy of the weighted, connected graph $G$ as:

$$\mathcal{H}^1(G) = -\sum_{u \in V} \frac{d_u}{\text{vol}(G)} \log_2 \frac{d_u}{\text{vol}(G)}, \tag{3}$$

where $\text{vol}(G)$ represents the total weighted degree of $G$. A higher $\mathcal{H}^1(G)$ indicates a more complex and less confidently represented region within the knowledge graph. By formulating SE as the exploration reward in MCTS, we enable the search algorithm to prioritize paths traversing maximally uncertain knowledge structures, thereby efficiently exposing the model's systemic weaknesses.

## 3.2   MCTS for Knowledge Deficiency Detection

Given the SE-based reward function, we employ MCTS to explore the KG and identify potential knowledge deficiency paths in the model. We define the initial state $s_0$ as the starting node for traversing the KG, where a set of seed entities from (Soman et al., 2024) is selected. KG triplets are incrementally incorporated into the knowledge paths until the maximum search depth $T$ is reached. This process enhances the LLM's awareness of its knowledge deficiencies by maximizing the expected reward, which emphasizes the uncertainty associated with these deficiencies.

**Node Selection.**    The objective of this stage is to identify and prioritize KG entities that are likely to expose the LLM's knowledge deficiencies, as shown in Figure 1a. Formally, at state $s_t$, the LLM agent reaches entity node $u_t$ of the KG, and the MCTS process choose from $\mathcal{A} = \{a_1, a_2, \ldots, a_m\}$, representing the relation edges $\rho_{t+1}$ that connect the current entity $u_t$ to its neighbors $\mathcal{N}(u_t)$. It is guided by two key variables: $Q(s_t, a)$, the cumulative value of taking action $a$ in state $s_t$, and $N(s_t)$, the visitation frequency of state $s_t$. Heuristically, $Q(s_t, a)$ guides exploitation by favoring actions with historically high rewards, while $N(s_t)$ encourages exploration of under-visited states. We integrate these complementary objectives using the PUCT algorithm (Rosin, 2011), which selects the next state as:

$$s_{t+1}^* = \arg\max_{s_t} \left[ Q(s_t, a) + c_{\text{puct}} \cdot P(a \mid s_t) \frac{\sqrt{N(s_t)}}{1 + N(s_t, a)} \right], \tag{4}$$

where $P(a|s_t)$ denotes the prior probability of selecting action $a$ given state $s_t$. In this way, an additional triplet $\tau$ is incorporated into the knowledge path $\mathcal{P}$.

**Path Expansion.** Expansion occurs when a leaf node is reached during the selection phase, enabling the integration of new states and the assessment of immediate rewards. Upon reaching a leaf node, it is expanded by selecting all possible relation action from leaf node, where each action $a$ represents a transition from the current entity state $s_t$ to a new entity state $s_{t+1}$ in $\mathcal{N}(s_t)$, as shown in Figure 1b. These unexplored entities $\mathcal{N}(s_t)$ are then added as leaf nodes to the search tree. The immediate reward function $r(s_t, a)$ quantifies the advantage of each action $a \in \mathcal{A}$ available at state $s_t$.

$$
\begin{aligned}
r_{t+1} = r(s_t, a) = I(s_t, a, s_{t+1}) = -\log_2 \frac{d_{s_{t+1}}}{\text{vol}(G)}, \\
V(s_t) = r_{t+1} + \gamma V(s_{t+1}) = \sum_{k=0}^{T-k-1} \gamma^k r_{t+k+1},
\end{aligned}
\tag{5}
$$

where $\gamma$ is the discount factor for future state values $V(\cdot)$ and $T$ is the depth of the MCTS search space. To accommodate scenarios with limited decision steps and stable reward distributions, we eliminate the discount factor and instead compute the average of future immediate reward values, as formalized in Equation 6.

**Reward Estimation.** A simulation shown in Figure 1c is run from the new expanded node $s_t$ by making random relation actions until a terminal state is reached. The newly expanded nodes are evaluated using an evaluation function integrating future rewards, state relevance, and actual outcomes. In this paper, we propose a novel intrinsic reward mechanism to address the limitation of Shannon entropy in handling structured data. To overcome this challenge, we define one-dimensional structural entropy as an intrinsic reward for effective exploration:

$$
\begin{aligned}
V(s_t) = H(\mathcal{P}) = \mathbb{E}\left[ \sum_{k=0}^{T-k-1} r_{t+k+1} \,\middle|\, s_t \right] \\
\approx \mathcal{H}^1(\mathcal{G}) = -\sum_{s_t \in \mathcal{P}} \frac{d_{s_t}}{\text{vol}(\mathcal{G})} \log_2 \frac{d_{s_t}}{\text{vol}(\mathcal{G})},
\end{aligned}
\tag{6}
$$

where $\mathcal{P} = \{s_t, s_{t+1}, \cdots, s_T\}$ denote the selection trajectory of $t$-th iteration, which ends at the terminal state $s_T$ after one complete simulation. For simplicity, the notation omits the relationships $\mathcal{A}$ between states. Specifically, $\mathcal{G}$ is a subgraph of the knowledge graph $G$, representing a given search space, and we utilize the structural entropy on this subgraph to approximate the state value.

**Backpropagation.** We update the statistics of each state in the tree that was traversed during the selection stage. Specifically, the back propagation process updates the value estimates and visit counts of all ancestor nodes along the trajectory $\mathcal{P}$ as shown in Figure 1d, ensuring leaf node evaluation informs higher-level decision-making. The updated rules are as follows:

$$
\begin{aligned}
N(s_t) \leftarrow N(s_t) + 1, \\
Q(s_t, a) \leftarrow \frac{1}{N(s_t, a)} \sum_{i=1}^{N(s_t)} \mathbb{I}_i(s_t, a) V_i(s_t),
\end{aligned}
\tag{7}
$$

where $N(s_t, a)$ is the number of times relation action $a$ has been selected from state $s_t$, $N(s_t)$ is the number of times a simulation has been run from state $s_t$, and $\mathbb{I}_i(s_t, a)$ is 1 if relation action $a$ was selected from state $s_t$ on the $i$-th simulation run from state $s_t$, or 0 otherwise.

### 3.3 Deficiency Knowledge Synthesis and Repair

As shown in Figure 1f to 1h, our framework leverages the trajectories with the highest SE values obtained via MCTS to guide synthetic data generation. Specifically, we prompt the LLM agent to synthesize a set of QA pairs based on the identified knowledge path on which the LLM shows high uncertainty, so that the knowledge deficiency of the LLM can be remedied by training on these QA pairs. Formally, as shown in Figures 5 and 6, given a trajectory $\mathcal{P} = \{s_1, s_2, \ldots, s_T\}$, the prompt instructs the LLM to generate a question that focuses on $\mathcal{P}$ and an answer that logically explains on the relationship $\rho_{t+1}$ between $s_t$ and its neighboring entities $\mathcal{N}(s_t)$ in $\mathcal{P}$. So that the synthesized QA pair can adhere to the underlying knowledge about the knowledge path and remedy

the knowledge deficiency of the LLM. Furthermore, to maintain high data quality, we implement a multi-tiered evaluation mechanism that includes both heuristic rules and LLM-based judgments. Our quality standards encompass: *Format Consistency:* The generated QA pairs must strictly adhere to the predefined prompt template, ensuring that the structure, punctuation, and length conform to our specifications. This guarantees that the synthesized data maintains a uniform format that facilitates downstream processing. *Logical Coherence:* The QA pairs must exhibit clear and rational reasoning. The answer should provide a logically consistent explanation that reflects the relationships and context derived from the knowledge trajectory, ensuring that the data effectively captures and addresses the identified knowledge deficiencies. *Hallucination Avoidance:* The generated content must be grounded in the input trajectory. Specifically, all entities and facts mentioned in the QA pair must originate exclusively from the given trajectory, preventing the introduction of extraneous or unsupported information that could undermine the model's reliability. Data samples that do not meet these criteria are filtered out through our evaluation mechanism A.1, thereby ensuring that only high-quality synthetic data is used to remediate the LLM's knowledge gaps.

The training process can be divided into two stages: First, a knowledge injection stage, that aims to enrich the LLMs with deficiency medical knowledge $D_K$. Second, a medical instruction tuning stage, that tailors the model to align with the medical QA domain. (see Appendix A.3 for details).

# 4    Experiments

We conduct experiments on the knowledge-intensive *medical domain* to investigate the following research questions (RQs): **RQ1**: Can the proposed SENATOR framework effectively repair the knowledge deficiencies of existing LLMs? **RQ2**: How do different components of our proposed framework impact the performance of LLMs? **RQ3**: Does the synthetic data successfully incorporate knowledge that lies beyond the distribution of the pretraining corpus? **RQ4**: What is the scaling regularity of synthetic data on model performance?

## 4.1    Experimental Settings

**Language Models** We evaluate our methodology on two categories of LLMs: 1) General LLMs: We employ Llama-3-8B and Qwen2-7B as base models to examine the effectiveness of our approach and include Baichuan2 and Llama-2 for comparison. 2) Medical LLMs: Med-Alpaca (Han et al., 2023): Fine-tuned on LLaMA-13B with medical instruction data from Alpaca (Han et al., 2023), specifically designed for medical dialogues and question-answering tasks. PMC-LLaMA (Wu et al., 2024): Enhanced with biomedical knowledge from 4.8 million academic papers and 30,000 medical books, followed by medical-specific instruction tuning on LLaMA-13B. HuatuoGPT-II (Chen et al., 2023a): Built on Baichuan (Yang et al., 2023a), fine-tuned with distilled ChatGPT data and real-world medical data from doctors.

**Datasets** Our instruction tuning data $D_I$, which contains 514k samples, is derived from Wu et al. (2024) to align with the medical domain. It's widely used in the medical field for its large scale and comprehensive coverage of medical knowledge. We evaluate our approach on five standard medical benchmarks: 1) **MedQA** (Jin et al., 2021): Multiple-choice questions from the USMLE assessing medical understanding and reasoning. 2) **MedMCQA** (Pal et al., 2022): Over 194K questions from AIIMS exams covering 2,400 topics across 21 subjects. 3) **PubMedQA** (Jin et al., 2019): A biomedical QA dataset from PubMed abstracts with 1K expert-annotated and 211K generated QA instances, designed to test comprehension and reasoning in biomedical research. 4) **GPQA** (Rein et al., 2023): A high-difficulty multiple-choice dataset validated by experts in biology, physics, and chemistry, focusing on interdisciplinary knowledge and reasoning. 5) **MMLU** (Hendrycks et al., 2020): A comprehensive benchmark covering 57 tasks for evaluating large language models.

**Knowledge Graph** We conduct experiments based on the SPOKE knowledge graph (Morris et al., 2023) due to its comprehensiveness on biological and medical knowledge, which contains over 42 million nodes of 28 different types and 160 million edges of 91 types, constructed by integrating information from 41 different biomedical databases. In this paper, the initial seed entities for MCTS are common disease entities in SPOKE, sourced from Soman et al. (2024).

Table 1: Main Results on Medical Benchmarks in the Zero-shot Setting. $\Delta$ represents the relative change in performance when using our synthetic data generated by SENATOR compared to the corresponding backbone model. "w/" denote "with" and IT represents instruciton tuning data.

| Model | MedQA | MedMCQA | PubMedQA | GPQA | | Avg. |
| --- | --- | --- | --- | --- | --- | --- |
| | | | | Genetics | Molecular Biology | |
| Human (pass) | 50.0 | – | 60.0 | 43.2 | | – |
| Human (expert) | 87.0 | 90.0 | 78.0 | 66.7 | | 80.43 |
| *Medical LLMs* | | | | | | |
| Chat-Doctor (7B) | 33.93 | 31.10 | 54.3 | – | – | – |
| Med-Alpaca (13B) | 30.85 | 31.13 | 53.2 | 10.0 | 15.43 | 28.12 |
| HuatuoGPT-II (7B) | 41.13 | 41.87 | 54.2 | 22.5 | 21.60 | 36.26 |
| HuatuoGPT-II (13B) | 45.72 | 38.75 | 51.6 | 20.0 | 27.78 | 36.77 |
| PMC-LLaMA (13B) | 50.67 | 50.18 | 59.8 | 15.0 | 27.16 | 40.56 |
| *General LLMs* | | | | | | |
| Baichuan2-7B | 34.56 | 35.12 | 60.2 | 20.0 | 20.99 | 34.17 |
| Baichuan2-13B | 43.60 | 39.25 | 50.7 | 27.5 | 30.86 | 38.38 |
| Llama-2-7B | 30.95 | 28.85 | 60.8 | 25.0 | 17.28 | 32.58 |
| Llama-2-13B | 31.26 | 29.00 | 62.2 | 35.0 | 20.99 | 35.69 |
| Llama-3-8B | 55.54 | 52.21 | 54.8 | 20.0 | 29.01 | 42.31 |
| w/ instruction tuning | 54.36 | 50.08 | 56.6 | 25.0 | 25.93 | 42.39 |
| w/ synthetic data + IT | 58.29 | 53.60 | 64.8 | 27.5 | 32.72 | 47.38 |
| $\Delta$ promotion | +4.95% | +2.66% | +18.25% | +37.50% | +12.79% | +11.98% |
| Qwen2-7B | 54.67 | 53.41 | 64.6 | 32.5 | 36.42 | 48.32 |
| w/ instruction tuning | 59.07 | 59.77 | 61.2 | 22.5 | 35.80 | 47.67 |
| w/ synthetic data + IT | 59.70 | 60.70 | 63.2 | 40.0 | 40.12 | 52.74 |
| $\Delta$ promotion | +9.20% | +13.65% | -2.17% | +26.08% | +10.16% | +9.15% |

## 4.2 Main Results (RQ1)

Table 1 presents the performance of our approach and baseline models across four medical benchmarks. From this, we observe that (1) Through continuous pretraining on medical corpora, previous medical domain LLMs such as PMC-LLaMA could achieve ordinary-human-level performance on certain benchmarks. For example, **PMC-LLaMA employs approximately 514k samples, 79 billion tokens of medical data** to achieve performances close to such as MedQA and PubMedQA. However, its performance on genetics-related subset of GPQA still shows a substantial gap with human-level, indicating significant knowledge deficiency. (2) In contrast, our proposed SENATOR framework demonstrates its effectiveness in finding knowledge deficiencies to efficiently adapt LLMs to the medical domain. When applied to Llama-3-8B and Qwen2-7B, the SENATOR framework uses a much smaller amount of synthetic data **(26k samples, 0.8 million tokens and 128k samples, 3.6 million tokens, respectively)** to remedy the targeted knowledge areas, and improve the performance on corresponding benchmarks. For instance, the SENATOR optimizes the Qwen2 model attains an accuracy of 40% on the Genetics component of GPQA, demonstrating that supplementing missing domain-specific data can substantially enhance performance. Overall, on the four medical domain-related benchmarks, on average, the SENATOR framework improves the performance of Llama-3-8B and Qwen2-7B for 11.98% and 9.15%, respectively. This shows the effectiveness and generality of our approach in comprehensively detecting and remedying the domain-related knowledge for different LLMs. In the following paragraphs (RQ2 and RQ3), we demonstrate that the improvement stems from SENATOR's ability to effectively detect the knowledge deficiencies by synthesizing data beyond the original pretraining corpus, expanding its coverage, and optimizing its distribution.

## 4.3 Ablation Study (RQ2)

To validate the efficacy of SENATOR, we conduct ablation studies comparing three configurations: (1) base models, (2) models fine-tuned solely with general domain instruction data $D_I$, and (3) models

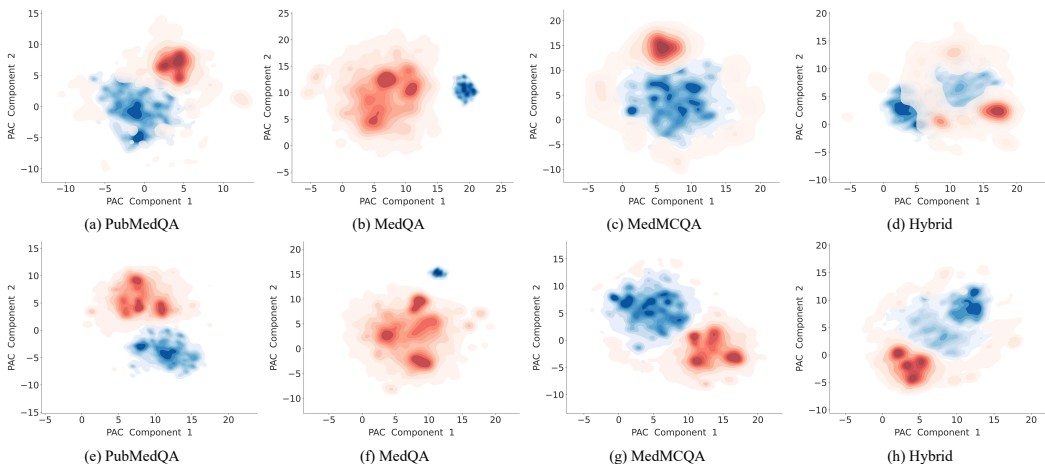

Figure 2: Distribution of Pretraining Corpus vs. Synthetic Data. In (a)-(d), blue regions represent the medical pretraining corpus (PubMedQA, MedQA, MedMCQA, and their hybrid), red regions show synthetic data generated by Llama-3. In (e)-(h), red regions indicate synthetic data produced by Qwen2. Darker areas reflect higher concentrations of data points, lighter areas vice versa.

trained with both instructions and synthesized data. As shown in Table 1, SFT on general domain instructions alone yields marginal improvements or even performance degradation (Llama-3-8B: 42.31 → 42.39; Qwen2-7B: 48.32 → 47.67). This suggests that the general domain instructions struggle to alleviate the intrinsic knowledge gaps in general-domain LLMs for the specialized medical domain, and constructing more general domain instructions would inevitably be inefficient. In contrast, incorporating synthetic data leads to a significant improvement. For Llama-3-8B, additional synthesized data make average performance improvements of 5.07, with particularly significant gains in underrepresented domains: +7.5 points in GPQA Genetics and +3.71 points in Molecular Biology. Similarly, Qwen2-7B attains 40.0% accuracy in GPQA Genetics (7.5-point increase) and 40.12% in Molecular Biology (3.7-point gain). These results indicate that performance improvement is brought by synthesizing data from detecting the deficiency of LLMs instead of simply enlarging the size of existing instruction data, and **a deficiency-oriented synthetic data generation strategy** would be a more efficient method for expanding knowledge of LLMs, suggesting a way towards "new fuel" (PwC Australia, 2023) for enriching the existing corpus and empowering future LLMs.

## 4.4 Analysis for Distribution of Synthesized Data (RQ3)

To examine if our approach can generate synthetic data beyond the original pretraining distribution and address the knowledge deficiency of LLMs, we visualize the distribution of both the original pretraining data, which is sourced from the training sets of PubMedQA, MedQA, and MedMCQA, and the synthetic data. This visualization is achieved by first projecting data into a unified semantic space using 2D UMAP (McInnes et al., 2018) and obtaining their distribution using kernel density estimation (KDE) (Rosenblat, 1956; Parzen, 1962). From Figure 2 we can observe: **1) Expanded Coverage by synthetic data:** Figures 2a to 2h reveal that the red area (representing synthetic data) encircles the blue area (pretraining data), indicating that the synthetic data effectively broadens the coverage of the pretraining data. Additionally, Figure 2b and 2f display smaller blue regions, indicating that the distribution of synthesized data is much broader than the

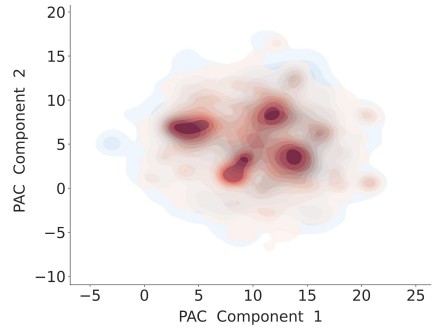

Figure 3: Distribution of Data Generated by Llama-3 (red) and Qwen2 (blue).

pretraining data available for MedQA. **2) Distribution Overlap:** In Figure 2d, the synthetic data

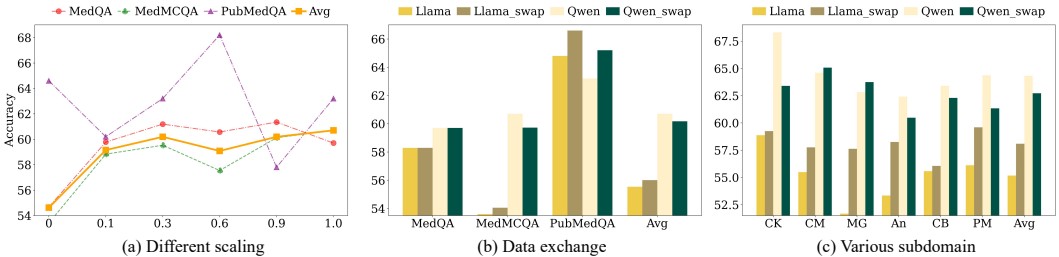

Figure 4: Performance differences for various data compositions.

shows a high degree of overlap with the overall pretraining data. We hypothesize that this may be due to Llama-3's relatively weaker grasp of pretraining knowledge compared to Qwen2, causing SENATOR to collect information that Llama-3 did not consolidate well during pretraining. **3) Topic-Specific Differences:** Compared to Figure 2a, Figure 2e exhibits an opposite trend. Accordingly, as indicated in Table 1, Qwen2 demonstrates a higher performance on PubMedQA. This is likely because Qwen2 demonstrated a stronger mastery of PubMedQA during pretraining (Yang et al., 2024a), leading SENATOR to explore that topic distribution to a lesser extent during the defect detection phase. **4) Global Trends and Localized Discrepancies:** The analysis of synthetic data distributions generated by Llama-3 and Qwen2 (Figure 3) shows substantial overlap in high-density areas, indicating that both models have a roughly similar pattern (may also share with more LLMs) in knowledge deficiency about the medical domain. This is because of the similarity in the distribution of the pretraining corpus (Lee et al., 2022; Yauney et al., 2023). Such similarity indicates the necessity of systematically reviewing the deficiencies of present LLMs to find common knowledge blind spots in the pretraining corpus, and synthesizing data to complement them. However, there still exist differences in certain locations, suggesting model-specific knowledge deficiencies. This suggests the effectiveness of our approach in targeting model-specific knowledge deficiencies.

## 4.5 Analysis of Synthetic Data Scaling (RQ4)

To explore how the amount of synthetic data affects model repair, we integrate different proportions of synthetic data into the SFT stage, as depicted in Figure 4a. We observe an upward trend in overall performance, calculated as a weighted average based on dataset sizes, with increasing synthetic data proportions. This indicates that, when the instruction-aligned data $D_I$ is fixed, expanding the synthetic data enhances model performance. As more synthetic data is used, more LLM knowledge deficiencies can be identified and addressed, thereby improving the model's performance. This highlights the potential of our method to effectively boost model performance by targeting and synthesizing data to fill specific knowledge gaps. Due to the limitation in computation resources, in this paper, for the two base LLMs, Llama and Qwen, we synthesize 26k and 128k data entries, respectively. In future work, we will explore integrating diverse knowledge across more domains to further enhance model performance. Additionally, we compare two settings: the default setting (SENATOR), where each model is fine-tuned using data synthesized using its own detected deficiencies, and the swap setting, where a model is trained with data synthesized using deficiencies of another model, for example, synthetic data produced by Llama-3 is used for SFT of Qwen2, and vice versa. As shown in Figure 4b and 4c, SENATOR demonstrates effective deficiency correction even under the swap setting. This could be brought by the similarities between the pretraining corpus of different LLMs, which can lead to similar knowledge deficiencies. This finding not only reinforces the potential of our synthetic data as a valuable supplement to human-written corpora, but also highlights the pressing need for efficient and comprehensive strategies to detect and repair knowledge deficiencies in LLMs.

## 5 Conclusion

In this paper, we introduce SENATOR, an innovative framework that utilizes structural entropy and knowledge graphs to detect and repair knowledge deficiencies in LLMs. By employing MCTS within the knowledge space, SENATOR effectively identifies areas where the model's understanding is deficient. Leveraging the SENATOR agent, we direct the synthetic data generation process to

specifically target these deficiencies. Our experiments on medical benchmarks reveal significant performance improvements when models like Llama-3 and Qwen2 are fine-tuned with the synthetic dataset. These results highlight that a deficiency-oriented synthetic data generation strategy represents a highly efficient and sustainable method for expanding knowledge, positioning it as the "new fuel" of modern AI.

## 6  Acknowledgements

We thank the support of the National Science and Technology Major Project (2022ZD0116301), the General Program of the National Natural Science Foundation of China (62576021) and the Youth Fund of the National Natural Science Foundation of China (62406040).

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

# A  Technical Appendices and Supplementary Material

## A.1  Prompts for Synthetic Data Generation Stage

This section introduces the prompts (Figure 5 and 7) defined in our synthetic data generation phase, including the question-answer paris generation prompt, and the evaluation prompt. And Figure 6 shows a specific example generated by SENATOR using the generation prompt.

---

**Synthetic Data Generator (Step 1)**

For given facts, generate a question and its corresponding answer. The question should be designed to inquire about the relationship or classification described in the triples, and the answer should be an entity mentioned in the provided facts.
Facts:
Disease <Thyroid Gland Mucoepidermoid Carcinoma> is a type of disease <thyroid gland carcinoma>.
Compound <Liothyronine> treats disease <thyroid gland carcinoma>.
Question: What compound can be used to treat Thyroid Gland Mucoepidermoid Carcinoma?
Answer: Liothyronine.

Facts:
Disease <thyroid gland carcinoma> resembles disease <ganglioneuroma>
Disease <ganglioneuroma> presents Symptom <Diarrhea>
Question: What symptom is associated with the disease that resembles thyroid gland carcinoma?
Answer: Diarrhea.

Facts:
Disease <head and neck cancer> resembles <thyroid gland carcinoma>.
Disease <head and neck cancer> presents Symptom <Dysphonia>.
Disease <head and neck cancer> presents Symptom <Neck Pain>.
Disease <thyroid gland carcinoma> presents Symptom <Dysphonia>.
Disease <thyroid gland carcinoma> presents Symptom <Neck Pain>.
Compound <Paclitaxel> treats disease <head and neck cancer>.
Question: What disease is similar to thyroid gland carcinoma, with Symptom Dysphonia and Neck Pain.
Answer: Head and neck cancer.

---

Figure 5: Example prompt for the synthetic data generation stage of SENATOR.

---

**A Smaple Generated by SENATOR**

{generation prompt}
# ***Input: Maximum Structual Entropy Trajectory by SENATOR***
Disease <hyperphosphatemia> contraindicates the use of compound <Retinol>,
Compound <Retinol> is contained in food <hickory nut>,
Food <hickory nut> contains compound <Tryptophan>,
Compound <Tryptophan> is contained in food <cow milk (liquid)>

# ***Output: QA Samples generated by the LLMs***
Question: Which compound, present in both hickory nut and cow milk (liquid), is safe for consumption by an individual with hyperphosphatemia?
Answer: Tryptophan.

---

Figure 6: A specific example generated by SENATOR.

## A.2  Prompts for the SFT Evaluation Stage

This section introduces the evaluation prompt (Figure 8) used after model knowledge repair, as shown in Figure 1h, designed to align the model's output answers with the desired format in the medical domain. Specifically, we employ a zero-shot setting in our evaluation to reduce the model's sensitivity bias to few-shot examples.



**Sample Evaluation Scorer (Step 2)**

Your task is to evaluate the given QA Pairs with Evidences based on the following criteria.
The criteria should include three parts:

Format: Verify the question is complete (i.e., not truncated) and can be answered with a single, clear answer. Check the answer is complete (i.e., not truncated) and is presented in a single entity or a concise subject-predicate-object statement.

Logic: Confirm that there is a clear, derivable logical connection between the question and the answer based on the provided Evidences.

Hallucination: This examines whether the entities involved in the question and answer exist within the provided Evidences. It determines if additional information beyond the given Evidences was used to construct the samples.

For each QA sample, analyze whether it meets the above criteria. If the sample satisfies all criteria, output "Correct". Otherwise, output one of the error types that best describes the issue with the sample: Format, Logic, or Hallucination.



Figure 7: Example prompt for the sample filtering stage of SENATOR.



**Eval Prompt for Medical Datasets**

*# Instruction:*
Directly answer the best option or Directly answer yes/no/maybe:

*# Example (PubMedQA):*
# Abstract: Electrical neurostimulation has traditionally been limited to the use of charge-balanced waveforms. Charge-imbalanced and monophasic waveforms are not used to deliver clinical therapy, because it is believed that these stimulation paradigms may generate noxious electrochemical species that cause tissue damage. In this study, we investigated the dissolution of platinum as one of such irreversible reactions over a range of charge densities up to 160 μC cm. We observed that platinum dissolution decreased during charge-imbalanced and monophasic stimulation when compared to charge-balanced waveforms
# Question: Does electrical neurostimulation with imbalanced waveform mitigate dissolution of platinum electrodes?

*# Example (MedQA)*
# A 3-month-old baby died suddenly at night while asleep. His mother noticed that he had died only after she awoke in the morning. No cause of death was determined based on the autopsy. Which of the following precautions could have prevented the death of the baby?
# A. Placing the infant in a supine position on a firm mattress while sleeping.
# B. Keeping the infant covered and maintaining a high room temperature.
# C. Application of a device to maintain the sleeping position.
# D. Avoiding pacifier use during sleep.

*# Example (MedMCQA):*
# Which vitamin is supplied from only animal source:
# A. Vitamin C   B. Vitamin B7   C. Vitamin B12   D. Vitamin D



Figure 8: Example prompt for the evaluation on medical datasets, where the "#" symbol denotes comments illustrating how a specific data sample is combined with an instruction for zero-shot prompting.

## A.3 Supervised fine-tuning hyperparameters

We use cross-entropy for supervised fine-tuning. Table 2 presents the hyperparameters utilized for SFT of LLMs within the SENATOR framework. As shown in Table 2, the settings applied to Llama-3-8B are identical to those of Qwen2-7B. Moreover, all experiments conducted in this paper have been performed using the same hyperparameter configuration.

Table 2: Model Training Parameters in SENATOR

| Model | Learning Rate | Weight Decay | Warmup Step | Batch Size | Epoch | Maximum Sequence Length |
|---|---|---|---|---|---|---|
| Llama-3-8B | 9.65e-6 | -1 | -1 | 1 | 3 | 1024 |
| Qwen2-7B | 9.65e-6 | -1 | -1 | 1 | 3 | 1024 |

## A.4 Data Filtering

While our framework demonstrates significant improvements over baseline methods, we acknowledge that the system remains imperfect. To systematically evaluate its limitations, we conduct a manual examination of 501 randomly sampled QA pairs from SENATOR outputs. The analysis revealed that 311 samples (62.08%) met our quality criteria for valid question-answer pairs. The remaining 190 error-containing samples (37.92%) exhibited the following error distribution: Formulaic errors (84 samples; 16.77%): Questions or answers with truncations, formatting inconsistencies, or multi-answer requirements. Logical errors (98 samples; 19.56%): Answers lacking evidential support from the provided knowledge triples. Hallucination errors (8 samples; 1.59%): Answers referencing entities absent in the supporting evidence. Notably, while our approach effectively mitigates hallucination errors through evidence grounding, generating logically consistent QA pairs remains challenging. This primarily stems from the base model's inherent limitations in performing multi-hop reasoning across knowledge path. Appendix A.7 illustrates representative examples of these error categories, demonstrating both the framework's capabilities and its current limitations. In order to improve data quality, we set up an additional data filtering module. For format problems, we use regularization to remove samples that do not meet specifications. For logical error types, we use LLMs to judge the logical consistency of QA pairs and evidences, and filter out unsatisfied samples.

## A.5 Impact of synthetic data on different medical subfields

Similar phenomena as shown in 4 can also be observed in different medical-related subdomains in the MMLU dataset, as shown in Figure 9. Our analysis on Qwen2 shows that without sythetic data generated by SENATOR (ratio = 0), performance is lowest. As synthetic data increases, sub-domain performance improves but with fluctuations. We attribute this to SENATOR's lack of entity type consideration during KG exploration, causing random data domains and non-uniform categories. Future work will focus on adding entity type constraints in MCTS search to explore domain specific knowledge deficiencies more precisely.

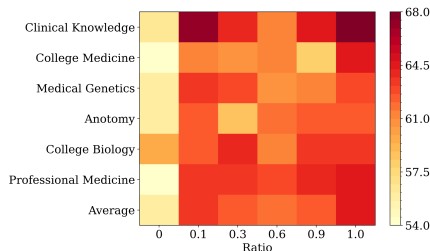

Figure 9: Performance across Different Ratios in MMLU Medical Aspects.

## A.6 Comparison with Latest Medical LLM Baselines

To provide a more comprehensive evaluation against recent state-of-the-art medical LLMs, we have added new baselines including BioMistral-7B (Labrak et al., 2024), Meditron-7B (Chen et al., 2023b), Llama-3-8B-UltraMedical (Zhang et al., 2024), and Qwen2-7B w/ SENATOR. The results are presented in Table 3 below.

Table 3: Model Performance on Medical QA Benchmarks

| Model | MedQA | MedMCQA | PubMedQA |
|---|---|---|---|
| BioMistral-7B | 44.93 | 42.17 | 56.4 |
| Meditron-7B | 30.40 | 31.22 | 61.6 |
| Llama-3-8B-UltraMedical | 56.75 | 53.75 | 52.12 |
| Llama-3-8B w/ SENATOR | 58.29 | 53.60 | 64.8 |
| Qwen2-7B w/ SENATOR | 59.70 | 60.70 | 63.2 |

## A.7    Case Stduy

Our framework SENATOR generates <evidence, question, answer> examples based on the SPOKE knowledge graph. These examples are categorized into four types: Correct, Formulaic errors, Logical errors, and Hallucination errors. Specific examples are illustrated in Figures 10 to 13.

---

Evidence:
Disease <hyperphosphatemia> contraindicates the use of compound <Retinol>,
Compound <Retinol> is contained in food <hickory nut>,
Food <hickory nut> contains compound <Tryptophan>,
Compound <Tryptophan> is contained in food <cow milk (liquid)>

Question:
What food contains the compound that is contraindicated in hyperphosphatemia?

Answer: Hickory nut

Comment: Correct

---

Figure 10: Correct Case.

---

Evidence:
Disease <primary ciliary dyskinesia 25> is a type of disease <primary ciliary dyskinesia>,
In genetics, disease <primary ciliary dyskinesia> associates with gene <MCIDAS>,
Gene <MCIDAS> downregulated in tissue <ectocervix>

Question:
In which tissue is gene MCIDAS upregulated?

Answer: Endometrium

Comment: Hallucination error

---

Figure 11: Hallucination Error Case.

## A.8    Details of the Instruction Tuning Dataset

**Medical Conversation Data**: the dataset includes approximately 100k instances from the ChatDoctor corpus, which contains diverse doctor-patient dialogues collected from real-world scenarios. To enhance instruction diversity and robustness, each prompt is expanded into multiple semantically equivalent forms using GPT-4.

**Medical Rationale Question Answering**: the dataset incorporates three major multiple-choice QA benchmarks: MedQA (10.2K examples), MedMCQA (183K), and PubMedQA (211K). These datasets evaluate the model's ability to reason over professional medical knowledge. Since many of these resources originally lacked detailed rationales, additional causal explanations were obtained by prompting ChatGPT, allowing the model to learn both the correct answer and the underlying reasoning.

Evidence:
Disease <acute necrotizing encephalitis> resembles disease <encephalomyelitis>,
Disease <encephalomyelitis> presents symptom <Myalgia>,
Symptom <Myalgia> can be caused by the side effect of compound <Diazepam>

Question:
What disease has a similar presentation to acute necrotizing encephalitis, with a symptom that can be treated by Diazepam?

Answer: Encephalomyelitis

Comment: Logical error

Figure 12: Logical Error Case.

Evidence:
Disease <otulipenia> is a type of disease <autosomal recessive disease>,
Disease <autosomal recessive disease> includes disease <spondyloepiphyseal dysplasia Kondo-Fu type>,
Disease <spondyloepiphyseal dysplasia Kondo-Fu type> presents symptom <Cataract>,
Symptom <Cataract> can be caused by the side effect of compound <Imatinib>

Question:
What is the type of disease that presents symptom Cataract, and what is the side effect of Imatinib?

Answer: Spondyloepiphyseal dysplasia Kondo-Fu type, Cataract

Comment: Formulaic error

Figure 13: Formulaic Error Case.

**Knowledge Graph–Driven Prompting**: Furthermore, two smaller datasets—LiveQA (635 examples) and MedicationQA (690 examples)—are included to provide real-world clinical questions and drug-related knowledge, respectively. Finally, the dataset includes 99K samples derived from the UMLS medical knowledge graph, covering both entity descriptions and inter-entity relationships. This component is particularly useful for aligning the model with structured biomedical ontologies.

Together, these seven resources offer a diverse and comprehensive instruction set $D_I$, enabling the model to generalize across conversational, inferential, and knowledge-based medical tasks. More detailed information can be found in the (Wu et al., 2024)

## B Limitations

While SENATOR demonstrates promising results in identifying and repairing knowledge deficiencies within LLMs, several limitations remain. First, our framework relies on an external human-curated knowledge graph (KG) to simulate a realistic environment in which the model can perform structured

exploration. This setup enables the LLM to iteratively discover and repair its knowledge gaps through self-improvement. However, such reliance on a high-quality, domain-specific KG may limit the framework's applicability in settings where such structured resources are incomplete or unavailable. In future work, we plan to explore ways to relax this dependency, such as constructing approximate KGs automatically from textual corpora or using retrieval-augmented methods to complement structural guidance.

Second, while the structural entropy-guided exploration effectively identifies knowledge deficiencies, the process of synthesizing data to repair these deficiencies can be further improved. The quality of synthetic data plays a crucial role in downstream model performance. However, this paper places greater emphasis on detecting and targeting knowledge gaps rather than exhaustively optimizing the data generation process. In our current implementation, we adopt prompt-based synthesis strategies for simplicity and reliability. In future work, we aim to incorporate more advanced techniques—such as instruction-tuned generation, controllable sampling to enhance the relevance, diversity, and factuality of the synthesized data.

## C  Broader Impacts

Our work on the SENATOR framework for detecting and repairing knowledge deficiencies in large language models through targeted synthetic data generation has both promising benefits and potential risks for society.

**Positive Impacts**

- **Improved Reliability in High-Stakes Domains:** By systematically identifying and closing knowledge gaps, SENATOR can make LLMs more accurate and trustworthy in domains such as medicine, law, and scientific research, where factual precision is critical for patient care, legal reasoning, and scientific discovery.
- **Democratization of Domain-Adapted Models:** Synthetic data alleviates the dependence on expensive, expert-annotated corpora, enabling smaller organizations, research labs, and underserved communities to fine-tune powerful LLMs for specialized tasks without prohibitive annotation costs.
- **Rapid Adaptation to Emerging Knowledge:** In fast-moving fields (e.g., novel pathogens, new regulations), synthetic data guided by up-to-date knowledge graphs can help models stay current, supporting timely decision-making and dissemination of accurate information.

**Negative Impacts**

- **Bias Amplification and Inaccuracy:** If the underlying knowledge graph or pretraining data contain biases or errors, synthetic data may inadvertently reinforce these issues. Models improved on such data could perpetuate harmful stereotypes or spread misinformation.
- **Misuse for Misinformation:** High-quality synthetic data generation techniques could be exploited to create convincingly false or misleading domain-specific content (e.g., fraudulent medical advice or fabricated legal precedents), posing risks to public trust and safety.
- **Overreliance on Synthetic Data:** An overconfidence in models fine-tuned primarily on synthetic data might obscure residual blind spots, leading users to place undue trust in automated systems without appropriate human oversight.
- **Privacy and Intellectual Property Concerns:** If knowledge graphs incorporate sensitive or proprietary information, there is potential for synthetic data to leak or replicate protected content, raising ethical and legal implications.

## D  Resource Requirement

We use 8 NVIDIA A100-40G GPUs to SFT Llama-3-8B and Qwen2-7B, and leverage 1-2 NVIDIA A100-40G GPUs for all the inference experiments.

Taking Qwen2-7B as an example, when using synthetic data to SFT Qwen2-7B for knowledge repair, the training time is about 30h on 8 NVIDIA A100-40G GPUs, and a total of 3 epochs are performed.

The inference time such as synthetic data generation stage and evaluation stage, measured in seconds per sample, is calculated on an NVIDIA A100 GPU with vllm acceleration (e.g. Qwen2-7B model, which demands at least two A100 GPUs for deployment)

