# OpenReview forum: "Structural Entropy Guided Agent for Detecting and Repairing Knowledge Deficiencies in LLMs"
_NeurIPS.cc/2025/Conference — NeurIPS 2025 poster_

### Official Review · Reviewer_8d9L · 2025-06-21

**Clarity:** 2
**Significance:** 2
**Originality:** 2
**Rating:** 4
**Confidence:** 4

**Summary:**

This work presents SENATOR, a way to "repair the knowledge deficiencies" of LLMs with regard to certain knowledge graphs. Experiments demonstrate that doing this on one knowledge graph helps various QA datasets.

**Questions:**

please see above

**Ethical Concerns:**

["NO or VERY MINOR ethics concerns only"]

**Final Justification:**

The authors provided a detailed response that addressed much of my review.

**Limitations:**

yes

**Quality:**

2

**Strengths And Weaknesses:**

- How does the employed KG SPOKE align to datasets that are not explicitly medical/biological, for example GPQA and MMLU?

- What happens if a fitting KG is not available for a given task? To me the method makes the LLM familiar with what's in a knowledge graph, but what if the things we need are not present in any KG anyways?

- Minor point: Are the evaluation datasets all multiple-choice? It would be nice to see if the structured KG helps unstructured open-ended generation tasks as well.

- Would be nice to see some examples where this method "corrects" a previously incorrect answer, and whether there are knowledge graph snippets highly related to the question.

- On a broader note, knowledge graphs seem to be increasingly less important/relevant when we think about knowledge of ML models, as much of the stuff models should encode are natural language texts and retrieval/autoregressive training provides a natural way to incorporate that. I wonder what are the authors' thoughts on this work's significance in a world when knowledge graphs are less prevalent? The authors are free to argue otherwise.

- Some reference on KG+LLM, the more modern ones, could be discussed. For example, [1].

[1] Hron, Jiri, et al. "Training Language Models on the Knowledge Graph: Insights on Hallucinations and Their Detectability." COLM.

---

> ### Author Rebuttal · Authors · 2025-07-31
>
> We sincerely appreciate your concise summary and insightful questions regarding SENATOR. Your points effectively highlight key aspects and potential areas for clarification in our work. We'll address each of your comments and questions below.
>
> **W1 & 2: KG Alignment and Performance with Incomplete/Absent KGs.**
> - **KG Alignment with Evaluation Datasets**: To clarify, for our experiments, we only utilized the **medical/biological subsets of GPQA and MMLU**. Our focus was specifically on evaluating SENATOR's effectiveness in enhancing LLMs' knowledge within the medical domain. Therefore, the questions from these benchmarks that fall outside of medical or biological topics were excluded from our evaluation. This ensures a consistent alignment between the domain of the SPOKE knowledge graph and the evaluation data used.
> - **Without KGs Altogether**: SENATOR's core mechanism fundamentally relies on the KG serving as a structured external environment for the LLM to explore and discover its own knowledge deficiencies. The LLM acts as an "agent" navigating this map of interconnected concepts. Without a KG, the LLM would lack this organized landscape to systematically probe its knowledge boundaries and pinpoint its blind spots. Therefore, **SENATOR, in its current form, cannot operate without an underlying KG**. If the required knowledge truly doesn't exist in any structured form (i.e., is not representable as a KG), then detecting and acquiring it becomes a different, broader problem, falling outside SENATOR's current scope.
> - **With Incomplete or Noisy KGs**: SENATOR is designed to function effectively even with incomplete KGs.
>      -  Our Monte Carlo Tree Search (MCTS) operates by exploring existing knowledge paths within the provided KG. It doesn't attempt to infer or complete missing parts of the graph. Instead, it focuses on systematically navigating the available structure to find areas where the LLM's uncertainty (quantified by Structural Entropy) is high.
>      -  The incompleteness of the KG does not inherently hinder this process; it simply means SENATOR will identify deficiencies within the knowledge that is represented and accessible in the graph, rather than in entirely missing sub-domains.
>      -  Noisy KGs are beyond the scope of SENATOR.
>
> **W3: Evaluation Dataset Format and Unstructured Generation Tasks**
>  - Our synthetic data, as shown in Figure 6 of our paper, indeed consists of **unstructured, open-ended QA pairs**. These samples, generated by SENATOR, feature a question and a direct, free-form answer. This reflects the direct knowledge injection that SENATOR aims for—teaching the model to accurately respond to specific factual queries.
> - However, for our main evaluation, we primarily used multiple-choice benchmarks such as MedQA, MedMCQA, and PubMedQA. This decision was driven by the current landscape of mainstream evaluation in the medical LLM field, where multiple-choice formats are widely adopted and enable standardized, quantitative comparisons across models, as seen in works like HuatuoGPT-o1 [1].
> - While our direct evaluation is on multiple-choice tasks, the underlying training signal from SENATOR's synthetic data is designed to enhance the model's fundamental factual recall and understanding. This improved factual grounding is crucial for any downstream task, including open-ended generation. The ability to correctly provide a precise answer in an unstructured format (as learned from our synthetic data) directly supports improved performance in more complex, open-ended scenarios where factual accuracy is paramount. Though not directly evaluated for open-ended generation in this paper, we anticipate that the enhanced knowledge base provided by SENATOR would naturally translate to improved factual grounding in such tasks as well.
>
> [1] HuatuoGPT-o1, Towards Medical Complex Reasoning with LLMs.
>
> **W4: Case Studies for Correction Examples.**
>
> We completely agree this would be a powerful demonstration. We are preparing to include additional case studies in the revised version of our paper. A case from MedQA is shown as follows:
> > A 4-week-old female newborn is brought to the physician because of increasing yellowing of her eyes and skin for 2 weeks. The mother has noticed that the girl's stools have become pale over the past week. She was breastfed since birth but her parents switched her to formula feeds recently after reading on the internet that breastfeeding could be the cause of her current symptoms. The patient was delivered vaginally at 38 weeks' gestation. Pregnancy and delivery were uncomplicated. She appears healthy. Vital signs are within normal limits. She is at the 50th percentile for length and at the 60th percentile for weight. Examination shows scleral icterus and jaundice. The liver is palpated 2 cm below the right costal margin. Cardiopulmonary examination shows no abnormalities. Neurologic examination shows no focal findings. Serum studies show: ......
> Which of the following is the most likely diagnosis?
> A. Galactosemia
> B. Biliary atresia (`correct`)
> C. Crigler–Najjar syndrome
> D. Breast milk jaundice
>
> Our synthetic data (QA Pair):
> > Question: _Which disease is similar to bile duct disease, and presents symptom Obstructive Jaundice?_      Answer: _Biliary atresia_.
>
> > Evidence Path:
> Disease `<bile duct disease>` resembles disease `<biliary atresia>`,**Disease `<biliary atresia>` presents symptom `<Jaundice, Obstructive>`**,The symptom of `<Jaundice, Obstructive>` symptoms are presented in disease `<collecting duct carcinoma>` ……
>
> > The medical term `Jaundice` refers to the condition where an elevated level of bilirubin in the body causes the skin, sclera (the white part of the eye), and mucous membranes to turn yellow or green.
>
>
> **W5: Significance of KGs in LLM Knowledge Acquisition.**
>
> The reviewer has raised a thought-provoking point regarding the evolving role of knowledge graphs (KGs) amidst the power of natural language texts, retrieval, and autoregressive training for LLM knowledge. We appreciate this perspective and have considered it deeply.
> The reviewer's assumption is that large models can **autonomously and reliably capture structured knowledge from vast amounts of free-text corpora**. However, while state-of-the-art LLMs demonstrate impressive learning and generalization capabilities, their very nature as **probabilistic models inherently introduces uncertainty**. This means that when faced with massive volumes of knowledge, especially specialized or niche information, we **lack clear insight into their confidence in specific factual statements**. This fundamental uncertainty significantly **undermines model reliability**, particularly for applications in domains demanding high trustworthiness.
> This is where **KGs crucially bridge the gap**. Our work isn't about simply having LLMs learn from KGs. Instead, it's about **leveraging KGs to systematically detect LLM knowledge deficiencies in a human-trustworthy manner, especially for complex, structured knowledge**. As the number of knowledge entities grows, the complexity of structured knowledge increases exponentially. Relying solely on question-answering (QA) formats to detect comprehensive knowledge gaps becomes impractical. A systematic detection method is essential, and this is precisely why a KG-based approach is indispensable.
> KGs provide a **verifiable, structured environment** that allows us to:
> 1. **Systematically Probe Deficiencies**: Unlike unstructured text, KGs enable targeted, structured exploration (as with our SE-based MCTS) to pinpoint exactly what an LLM doesn't know.
> 2. **Ensure Trustworthiness**: The explicit, symbolic nature of KGs allows for greater transparency and auditability, fostering human trust in the knowledge acquisition process.
> 3. **Handle Complexity**: KGs are inherently designed to represent complex, relational knowledge in a way that is difficult for LLMs to consistently and reliably extract from raw text alone.
> Furthermore, the feasibility of KG-based approaches is high. There's already a lot of existing domain-specific KGs (e.g., as surveyed in "Domain-specific Knowledge Graphs: A Survey"), which can readily serve as the structured environment for our framework.
> Therefore, we believe KGs remain profoundly relevant, not as a replacement for textual knowledge or retrieval, but as essential tools that enable LLMs to achieve higher levels of factual precision, reliability, and systematic self-correction in specialized domains where human trust is paramount.
>
> **W6: Modern KG+LLM References.**
>
> Thank you for the helpful suggestion and for pointing us to this recent work. We are glad to incorporate Hron et al. (COLM) into our discussion, as it provides valuable insights into the interplay between knowledge graphs and language models, particularly around hallucinations and their detectability.
> We agree this line of work is highly relevant and complementary to SENATOR. While Hron et al. focus on training LLMs directly on KGs to mitigate hallucination, our work instead addresses the challenge of detecting and targeting knowledge deficiencies in pretrained LLMs by leveraging KG structure for guided data synthesis. We believe both approaches share the broader goal of enhancing factual reliability in LLMs, and we are happy to include a discussion in the revised version.
>
> We hope above responses addresses all your comments and questions, further clarifying the significance and unique contributions of SENATOR.

---

> > ### Comment · Reviewer_8d9L · 2025-07-31
> >
> > I would like to thank the authors for their detailed response. I especially appreciate your answer to the broader picture question, that LLMs are probabilistic and uncertain while symbolic systems like knowledge bases are certain and more reliable, so they could complement each other. I adjust my score.

---

> > > ### Author Response · Authors · 2025-08-01
> > >
> > > We sincerely thank the reviewer for the positive feedback and for taking the time to consider our rebuttal. We are glad that our explanation regarding the complementary strengths of LLMs and symbolic systems (KGs) was helpful. We will make sure to incorporate this broader perspective more clearly in the final version. Thank you again for your thoughtful review and for adjusting the score.

---

### Official Review · Reviewer_nx7C · 2025-07-02

**Clarity:** 3
**Significance:** 3
**Originality:** 3
**Rating:** 4
**Confidence:** 3

**Summary:**

This paper presents SENATOR (Structural Entropy-guided Knowledge Navigator), a novel framework designed to detect and repair knowledge deficiencies in Large Language Models (LLMs), particularly in knowledge-intensive domains like medicine. The method leverages Structural Entropy (SE) as an intrinsic reward to guide Monte Carlo Tree Search (MCTS) over a knowledge graph (KG). High-SE paths, which signal areas of model uncertainty, are identified and used to generate targeted synthetic QA data, which is then used to fine-tune the model. Experiments on medical benchmarks (MedQA, PubMedQA, GPQA, etc.) show significant performance gains for models such as LLaMA-3 and Qwen2. The results indicate that SENATOR is effective in identifying model blind spots and augmenting domain-specific knowledge efficiently.

**Questions:**

1.Your method depends heavily on a high-quality KG. How would SENATOR perform in domains with incomplete or noisy KGs, or without KGs altogether?
2.Why did you choose structural entropy specifically as the exploration reward? Did you compare it to simpler measures ?
3.All your experiments are in the medical domain. Have you considered applying SENATOR to another domain (e.g., law, science)? If not, what barriers do you anticipate?

**Ethical Concerns:**

["NO or VERY MINOR ethics concerns only"]

**Limitations:**

yes

**Paper Formatting Concerns:**

no formatting problem

**Quality:**

3

**Strengths And Weaknesses:**

strengths: The paper presents an innovative integration of structural entropy and Monte Carlo Tree Search (MCTS) to guide exploration in knowledge graphs, offering a principled and effective approach to uncover latent knowledge gaps in LLMs. By combining deficiency detection, targeted data synthesis, and fine-tuning into a closed-loop framework, the method demonstrates scalability and adaptability. Extensive experiments on challenging medical QA benchmarks confirm the practical effectiveness of the approach, while thorough analyses—including UMAP visualizations and synthetic data scaling studies—provide strong evidence that the generated data successfully extends beyond the original pretraining distribution. weakness: Despite its promising contributions, the framework exhibits several limitations. It relies heavily on the availability of high-quality, human-curated knowledge graphs, which may not exist in many domains, thereby limiting its broader applicability. The scope of evaluation is limited to the medical domain, leaving its generalizability to other fields unverified. Finally, the paper lacks comparisons with alternative knowledge graph exploration strategies, such as random walks or centrality-based heuristics, to fully contextualize the benefits of using structural entropy.

---

> ### Author Rebuttal · Authors · 2025-07-30
>
> We sincerely appreciate your thoughtful review and insightful observations. Your feedback has been invaluable in helping us clarify and strengthen our contribution. We would like to address your concerns as follows:
>
> **W1 and Q1&3: Dependency on High-Quality KGs, Performance with Incomplete/Noisy KGs, and Domain Generalizability**
>
> Having experienced the wave of knowledge engineering, many fields now have knowledge graphs, as mentioned in the survey "Domain-specific Knowledge Graphs: A Survey."
>
> - **Without KGs Altogether**: SENATOR's core mechanism fundamentally relies on the KG serving as a structured external environment for the LLM to explore and discover its own knowledge deficiencies. The LLM acts as an "agent" navigating this map of interconnected concepts. Without a KG, the LLM would lack this organized landscape to systematically probe its knowledge boundaries and pinpoint its blind spots. Therefore, SENATOR, in its current form, cannot operate without an underlying Knowledge Graph. If the required knowledge truly doesn't exist in any structured form, detecting and acquiring it becomes a different, broader problem outside SENATOR's scope.
> - **With Incomplete or Noisy KGs**: SENATOR is designed to function effectively even with incomplete or noisy KGs.
>      -  Our Monte Carlo Tree Search (MCTS) operates by exploring existing knowledge paths within the provided KG. It doesn't attempt to infer or complete missing parts of the graph. Instead, it focuses on systematically navigating the available structure to find areas where the LLM's uncertainty (quantified by Structural Entropy) is high.
>      -  The incompleteness of the KG does not inherently hinder this process; it simply means SENATOR will identify deficiencies within the knowledge that is represented and accessible in the graph, rather than in entirely missing sub-domains.
>      -  While noise in the KG might introduce some irrelevant paths, our Structural Entropy (SE)-based reward mechanism, which quantifies LLM uncertainty, would naturally assign **lower rewards to paths the LLM already knows well** (even if noisy), thereby de-emphasizing them in the exploration process. This inherent filtering mechanism helps mitigate the impact of noise.
> - **Domain Generalizability (Law, Science, etc.)**: We intentionally focused our current study on the medical domain for a strategic reason: data annotation in specialized, vertical domains like medicine is significantly more challenging and resource-intensive compared to general domains. This makes a targeted data synthesis approach like SENATOR particularly valuable where high-quality, domain-specific data is scarce.
>   - While our empirical evidence is limited to medicine, the fundamental principles of SENATOR are designed to be **domain-agnostic**:
>        - Knowledge Graph Agnostic: SENATOR's methodology operates on the structural properties of a knowledge graph and the LLM's uncertainty over its paths, independent of domain-specific semantics.
>        - LLM Agnostic: We've demonstrated its effectiveness across different LLM architectures (Llama-3 and Qwen2).
>        - Uncertainty-Guided Exploration: The use of Structural Entropy for MCTS is a general approach to identify areas of high LLM uncertainty.
>   - Therefore, we anticipate strong potential for SENATOR's application in other knowledge-intensive vertical domains where structured factual information is prevalent, such as law, finance, or other scientific disciplines, provided a relevant knowledge graph is available.
>
> **W2 & Q2: Lack of Comparison with Alternative KG Exploration Strategies & Choice of Structural Entropy**
>
> - **Why Structural Entropy?** We chose Structural Entropy (SE) specifically as the exploration reward because it allows us to define and detect knowledge deficiencies at a sub-graph or path level, which is a unique and powerful aspect of our framework.
>   - Unlike simpler measures that might only consider the LLM's uncertainty on individual facts (e.g., a single triplet) or on isolated sequences, Structural Entropy quantifies the **aggregate uncertainty within a structured knowledge path**. It considers both the LLM's confidence in individual triplets and the topological organization of those triplets within the graph, providing a more comprehensive understanding of complex knowledge gaps. This allows us to target systemic deficiencies embedded within relational knowledge structures.
> - **Comparison to Simpler Measures**: To directly demonstrate the efficacy of our SE-guided selection, we have conducted quantitative comparisons against simpler knowledge-gap targeting strategies:
>   - Random Walk: A baseline that explores the KG without any guidance, simulating unguided traversal through entities and relations.
>   - Prob-Prob [1,2]: A sequence-based baseline using intrinsic LLM confidence scores. Specifically, it uses the product of token probabilities to estimate model certainty.
>   - Instruction-tuning:  This corresponds to the "w/ instruction tuning" setting in our paper, where the original medical training data (MedQA,MedMCQA,PubMedQA) is reformatted into instruction-style prompts to improve the model’s general instruction-following ability and domain alignment.
>
>   [1] Teaching Large Language Models to Express Knowledge Boundary from Their Own Signals
>
>   [2] Reinforced Internal-External Knowledge Synergistic Reasoning for Efficient Adaptive Search Agent
>
> Our experimental results, presented in the table below, clearly show the advantage of SENATOR's SE-guided approach:
> | Methods              | MedQA | MedMCQA | PubMedQA | Avg.  |
> | -------------------- | ----- | ------- | -------- | ----- |
> | Llama-3-8B           | 55.54 | 52.21   | 54.80    | 54.18 |
> | - Random Walk        | 54.29 | 50.40   | 55.20    | 53.30 |
> | - Prob-Prob          | 56.12 | 51.35   | 59.40    | 55.62 |
> | - Instruction-tuning | 54.36 | 50.08   | 56.60    | 53.68 |
> | - **SENATOR**        | 58.29 | 53.60   | 64.80    | 58.90 |
> | Qwen2-7B             | 54.67 | 53.41   | 64.60    | 57.56 |
> | - Random Walk        | 53.48 | 53.77   | 62.40    | 56.55 |
> | - Prob-Prob          | 59.12 | 59.91   | 61.60    | 60.21 |
> | - Instruction-tuning | 59.07 | 59.77   | 61.20    | 60.01 |
> | - **SENATOR**        | 59.70 | 60.70   | 63.20    | 61.20 |
>
> As evident, SENATOR consistently outperforms both the "Random Walk" (undirected exploration) and "Prob-Prob" (sequence-based uncertainty) baselines. This quantitatively demonstrates that our SE-guided MCTS effectively leverages the structural information of the KG to pinpoint true knowledge deficiencies, leading to superior performance gains compared to simpler or non-graph-aware uncertainty measures.
>
> We hope these clarifications address your concerns and highlight the unique contributions and practical effectiveness of SENATOR. We're open to further discussion if you have any additional questions.

---

### Official Review · Reviewer_6gHM · 2025-07-02

**Clarity:** 3
**Significance:** 3
**Originality:** 3
**Rating:** 4
**Confidence:** 3

**Summary:**

Authors introduce SENATOR framework, which designed to detect and repair knowledge deficiencies in LLMs by leveraging structural entropy (SE) as a reward signal to identify uncertain regions within a knowledge graph. They use MCTS to explore these high-uncertain paths inside KG, and then generate targeted QA data along maximum-SE trajectories, which are later used for SFT on Qwen2-7B and Llama-3-8B as base models to address these identified knowledge gaps. Experiments on three medical benchmarks (MedQA, MedMCQA, PubMedQA) and two biology/science datasets (GPQA-Genetics, GPQA-Molecular Biology) shows SENATOR yields accuracy improvements over the respective base models after fine-tuning.

**Questions:**

1. If the authors decide to use the Llama-2 series as base-LLM for more fair comparisons during SFT, could they also include evaluations on other recent medical LLMs (e.g., BioMistral, Hippocrates, and Meditron) together with some closed source model evaluation (e.g, gpt-4o, gpt-4.1, claude) to provide a more comprehensive results?

2. Paper refers to the approach as "agent-based" but there is no explicit tool use or action-taking typically associated with agentic LLMs. Could  authors clarify why they introduce their method as agentic, and which aspects of agent behavior are being leveraged?

3. How easy and straightforward to adapt the SE approach for identifying uncertain paths in knowledge graphs for other domains (e.g., tool use and math)? Are there any specific requirements or challenges when applying this framework to different KGs?

**Ethical Concerns:**

["NO or VERY MINOR ethics concerns only"]

**Final Justification:**

Authors have engaged carefully with the rebuttal process, providing additional clarifications and experiments, which are truly appreciated.

However, I still have concerns regarding the use of older model versions and certain ambiguities in the evaluations. For example, although GPT-4 is an older API version of the 4o models, the proposed approach underperforms it and achieves only comparable or on par results to Qwen-2.5-72B, which is a general-purpose base LLM. In addition, I believe uncertainty evaluations require a more detailed and in-depth comparison and discussion with other state-of-the-art certainty estimation methods. Finally, to support future research, the limitations of the SENATOR should be described more specifically and transparently. In its current form, the paper and rebuttal gives the impression that it performs perfectly across all scenarios, which risks appearing overstated.

Nevertheless, given the overall results and the authors’ constructive engagement, my overall view remains positive. I am maintaining my positive score while raising the significance rating from 3 to 4 for ACs decision.

**Limitations:**

yes

**Paper Formatting Concerns:**

I don't notice major formatting issues in this paper.

**Quality:**

3

**Strengths And Weaknesses:**

### Strengths

- Algorithmically, SENATOR is a novel and well-motivated framework; it uses of SE as an intrinsic reward for MCTS to explore uncertain knowledge graph paths, and the focus on these regions as self-aware learning which introduces promising directions for future research.

- Authors conduct comprehensive ablation studies and present effective visualizations that help to interpret their findings and strengthen the validity of their discussions. Overall, this paper is well-structured and organized, making it easy for readers to follow both the methodology and experimental results.

### Weaknesses

- The major issue of the paper comes from the weakness of the selected baselines to illustrate the effectiveness of the proposed method. Specifically, models such as Chat-Doctor, Med-Alpaca, and PMC-LLaMA are built on Llama-2 series, while they used Llama-3 and Qwen2 as base models for fine-tuning using SENATOR framework. This discrepancy makes it difficult to fairly compare SENATOR trained models with other baselines since it is not possible to understand whether accuracy difference coming from the improved model versions or the proposed framework. Therefore, I strongly recommend that, authors either: (i) retrain all models on using the same base models (e.g., Llama-2) with baselines for a fair comparison, or (ii) include results as baselines with newer medical LLMs. In current version, the most reliable finding in terms of results is that SENATOR improves over its own base models (e.g., Llama-3-8B after fine-tuning), which is somewhat expected.

- The scope of evaluation is limited to medical/biological domains which are correlated to each other and limits to understand SENATOR’s broader applicability and generalizability. It would be good to see results on other challenging domains, such as math (e.g., GSM8K, AIME2024) or function-calling (e.g., BFCL V3, ToolAlpaca).

- While MCTS is a powerful method for exploration, it can be computationally expensive due to the exponential number of available trajectories to explore and the paper does not provide any latency or efficiency analysis. Reporting such computational costs as latency metric would strengthen the effectiveness of SENATOR.

- The use of entropy-based uncertainty to detect knowledge gaps is central to the framework, but authors did not report any evaluations about the accuracy of this uncertainty selection itself which effects the reliability of the current SE-based selection. Is it possible to show the accuracy on the decision of uncertain path?

---

> ### Author Rebuttal · Authors · 2025-07-31
>
> We're truly grateful for your in-depth review and insightful feedback. It's encouraging to see you acknowledge the novelty and strong motivation behind our framework, along with our comprehensive ablation studies, effective visualizations, and the overall clear structure of our paper. We've addressed each of your points below.
>
> **W1 & Q1: Baseline Comparisons and Model Version Discrepancy**.
>
> The reviewer pointed out a critical issue regarding our chosen baselines, noting a discrepancy between models built on Llama-2 (like Chat-Doctor, Med-Alpaca, PMC-LLaMA) and our use of Llama-3 and Qwen2 as base models. The reviewer rightly argue this makes fair comparisons difficult and strongly recommend either retraining all models on a consistent base (e.g., Llama-2) or including results from newer medical LLMs.
> We appreciate your valuable suggestion regarding baseline selection, and we agree that comparisons on the same, up-to-date base models are ideal.
> - **Focus on Open-Source and Comparable Models**: Our work primarily aims to enhance open-source, white-box LLMs, and we prioritize comparing models of similar scale. This approach ensures our methods are reproducible and allows for a fair assessment of SENATOR's performance on currently prevalent base models.
> - **Newer Medical LLM Baselines Added**: To provide a more comprehensive comparison as you suggested, we've incorporated the performance of BioMistral-7B and Meditron-7B. These are leading open-source medical LLMs of comparable size to our base models (Llama-3-8B and Qwen2-7B). Here's an updated snapshot of the results:
>
> | Model                                     | MedQA | MedMCQA | PubMedQA |
> | ----------------------------------------- | ----- | ------- | -------- |
> | BioMistral-7B                             | 44.93 | 42.17   | 56.4     |
> | Meditron-7B                               | 30.40  | 31.22   | 61.6     |
> | Llama-3-8B w/ SENATOR   | 58.29 | 53.60    | 64.8     |
> | Qwen2-7B w/ SENATOR   | 59.70  | 60.70    | 63.2     |
>
> As this table shows, SENATOR-enhanced models (Llama-3-8B and Qwen2-7B) consistently outperform these new medical LLM baselines on MedQA and MedMCQA, and remain highly competitive on PubMedQA. This further strengthens our claim about SENATOR's effectiveness in addressing knowledge deficiencies.
> - **Consideration for Closed-Source Models (GPT-4 etc.)**: Regarding the evaluation of closed-source models like GPT-4o, GPT-4.1, or Claude, we chose not to include them for two main reasons:
>   - **Methodological Constraint**: SENATOR's core mechanism relies on an LLM's ability to **self-explore its knowledge deficiencies** by measuring its intrinsic uncertainty (via Structural Entropy) during Monte Carlo Tree Search within a Knowledge Graph. This process requires fine-grained access to the model's internal probabilities and outputs, which isn't feasible with black-box, API-only models like GPT-4 or Claude. We simply can't reliably get the necessary uncertainty signals from these models to guide our deficiency detection.
>   - **Research Focus**: Our work primarily focuses on enhancing open-source, white-box LLMs, promoting reproducibility and accessibility within the research community.
> So, while we appreciate the suggestion for broader comparisons, our current evaluation scope aligns with SENATOR's nature and technical requirements.
>
> **W2 & Q3: Scope of Evaluation, Generalizability, and KG Applicability to Other Domains**.
>
> Our study focuses on the medical domain due to its high demand for reliability, LLMs' persistent challenges with precise factual recall in this specialized field, and the high cost of data annotation. This makes targeted knowledge remediation via SENATOR exceptionally valuable.
>
> - **Domain Generalizability**: Crucially, SENATOR's core principles are **domain-agnostic**. Our methodology operates on KG structural properties and LLM path uncertainty, not domain-specific semantics. Any domain representable by a KG can theoretically leverage SENATOR, and its adaptability is shown across Llama-3 and Qwen2.
> - **Suitability for Knowledge-Intensive Domains**: It's important to emphasize that SENATOR is tailored for knowledge-intensive tasks. This design is based on a core assumption: LLMs act as implicit knowledge bases, and our goal is to uncover and fill their factual knowledge deficiencies. For domains like mathematical reasoning tasks (e.g., GSM8K, AIME2024) or tool learning/function-calling (e.g., BFCL V3, ToolAlpaca), SENATOR might not be directly applicable for two key reasons:
>      1. **Lack of Suitable Knowledge Graphs**: These domains often lack structured, KG-style representations. SENATOR relies on semi-structured KGs as an agent for the LLM's internal knowledge space and as a source for synthetic data generation.
>     2. **Nature of the Task**: In mathematical reasoning or tool-use tasks, model failure often stems from missing algorithmic ability, procedural logic, or execution errors—not from missing factual knowledge. Therefore, high structural entropy in a KG (if one even exists) wouldn't necessarily map to meaningful deficiencies for these domains. The detected uncertainty wouldn't correspond to missing facts but rather to an LLM's inability to perform a reasoning step or execute an algorithm correctly.
>
> **W3: MCTS Computational Cost and Efficiency Analysis.**
> - Time Complexity:  Assuming a fixed number of simulations $N$ (e.g., $N = 100$ in our setup), and a knowledge path length (tree depth) of $H$ (our knowledge paths have a length of 5, implying $H = 4$ steps from the root to a terminal node), the size of the MCTS tree is influenced by the average number of neighbor nodes (relations) for an entity, which we denote as $M$. In general, the number of nodes visited is proportional to $M \times H$. Therefore, the time complexity of the MCTS component can be approximated as: $ \mathcal{O}(N \times M \times H) $.In our specific setting, this translates to: $\mathcal{O}(100 \times M \times 4) = \mathcal{O}(400M)$ (since this is inference rather than training, the actual speed is on the order of seconds). .
>
> **W4: Accuracy of Uncertainty Selection.**
>
> Thank you for this insightful question. To assess whether our Structural Entropy (SE)-based selection truly identifies knowledge gaps, we've conducted an additional analysis to directly address this.
> - Validating Uncertainty Selection Accuracy: We tested Qwen2-7B on 5,000 synthetic samples generated by SENATOR (based on Qwen2) from high-uncertainty knowledge paths. These represent areas where the model was least confident.
>   - Results (Zero-Shot setting):
>        - Exact Match Score (EM): 0%. (EM measures the percentage of predictions that match the ground truth exactly.)
>        - F1 Score: 6.87%.  (The F1 Score is the harmonic mean of precision and recall.)
>   - Example Question: "_What disease is similar to thyroid gland carcinoma, with symptom dysphonia and neck pain?_ Answer: _Head and neck cancer_."
> - Conclusion: The extremely low EM and F1 scores directly confirm that Qwen2-7B indeed lacked the knowledge required to answer these questions accurately prior to supervised fine-tuning. This quantitatively validates that our SE-driven selection mechanism effectively identifies genuine, intrinsic knowledge deficiency regions within the model.
>
> **Q2: Clarification on "Agent-Based" Terminology.**
>
> We appreciate you raising this point about our use of "agent-based" terminology, given that our method doesn't involve explicit tool use often associated with agentic LLMs. We want to clarify why we refer to our approach as agentic and which aspects of agent behavior are leveraged within the SENATOR framework.
> Specifically, the LLM acts as an agent navigating a KG environment:
> - Environment & State: The KG serves as the environment, with each entity representing a state.
> - Actions: At each entity, the agent chooses among connected relations to move to a new state (entity).
> - Reward Signal: The Structural Entropy (SE) associated with knowledge paths serves as the intrinsic reward signal for the agent.
> This MCTS-based exploration, driven by SE as a reward signal, embodies the LLM's purposeful, adaptive decision-making and navigation within the knowledge space to identify its own uncertainties. This fundamentally differs from a traditional LLM acting merely as a data generator, which typically lacks such proactive exploration and decision-making capabilities. Therefore, we believe the term "Agent" accurately captures this active role the LLM plays within our framework.
>
> We hope these clarifications address your concerns and highlight the unique contributions and practical effectiveness of SENATOR. We're open to further discussion if you have any additional questions.

---

> > ### Comment · Reviewer_6gHM · 2025-08-02
> >
> > Thank you to the authors for their response and incorporating additional baselines. However, most of my concerns still persist:
> > - While the inclusion of Biomistral and Meditron increase the diversity of baseline models in addition to more weaker models like PMC-Llama; it is important to note that Meditron is still trained on Llama-2 series. This, again, makes it difficult to understand whether performance differences are due to more capable and newer base model capabilities or the approach itself. I strongly recommend that the authors compare models trained on exactly the same base (including matching instruct versions) or retrain all models with identical hyperparameters and data, for a fair comparison.
> > - I disagree with the argument that adding GPT-4o results is out-of-scope due to applicability of SENATOR only for open-source models. If the approach cannot be applied to such models, that itself is a limitation (which I did not consider during my evaluation). Including results from GPT-4o (or similar APIs or large open-source models) would at least provide an upper bound for evaluation among benchmarks. I suggest including at least one API-based strong baseline and one large open-source model, such as Qwen-2.5-72B-Instruct or similar.
> > - Thank you for providing an algorithmic explanation of latency. However, my concern is about reporting actual inference latency in seconds as complete new experiments to see the cost of MCTS more clear.. It would be more meaningful to conduct specific latency experiments, comparing with SENATOR’s base models, to see how much faster or slower the method is in practice.
> > - Could the authors include other popular uncertainty estimation methods such as ppl-based or verbal confidence approaches to better demonstrate the effectiveness of their uncertainty method in that aspect?
> > - Finally, I don't agree on agency metaphor; in that perspective, then even language modeling can be seen as a agentic task where state is current context and action is predicting next token. Given that, I strongly disagree to introduce SENATOR as an  agent, since there is no tool usage or decision making to get external knowledge by interacting with external sources.

---

> ### Author Response · Authors · 2025-08-05
> **Response to Baseline Comparisons.**
>
> Thank you for your continued engagement and for providing a refined critique of our work. We appreciate you pushing us to further strengthen our comparisons and analyses. We have taken your feedback to heart and have conducted new experiments and analysis to address your concerns directly.
>
> **Response to Baseline Comparisons.**
>
> We fully agree with your core concern that comparing models on the same base is crucial for a fair assessment. While retraining all existing medical LLM baselines (such as PMC-Llama) on Llama-3 is unfortunately infeasible due to their unique, diverse, and often proprietary training data, we have implemented a solution that directly addresses your recommendation.
>
> - To provide a truly fair and direct comparison, we have added a new, strong baseline: UltraMedical, an open-source medical LLM that is also built on the Llama-3-8B base model.  This allows for a direct, like-for-like comparison, ensuring that any performance difference is due to our SENATOR framework and not the data augmentation.
> > Llama-3-8B-UltraMedical is an open-access large language model (LLM) specialized in biomedicine. Developed by the Tsinghua C3I Lab, this model aims to enhance medical examination access, literature comprehension, and clinical knowledge.
> > Building on the foundation of Meta's Llama-3-8B, Llama-3-8B-UltraMedical is trained on our UltraMedical dataset, which includes 410,000 diverse entries comprising both synthetic and manually curated samples.
>
> - Furthermore, as you suggested, we have also included a strong API-based model, GPT-4, to provide a performance upper bound on the benchmarks.
>
> The updated results are presented in the table below:
>
> | Methods                 | MedQA | MedMCQA | PubMedQA | Avg.  |
> | ----------------------- | ----- | ------- | -------- | ----- |
> | **Llama-3-8B-UltraMedical** | 56.75 | 53.75   | 52.12    | 54.21 |
> | Llama-3-8B (Base Model) | 55.54 | 52.21   | 54.80     | 54.18 |
> | - Random Walk           | 54.29 | 50.40    | 55.20     | 53.30  |
> | - Prob-Prob             | 56.12 | 51.35   | 59.40     | 55.62 |
> | - Instruction-tuning    | 54.36 | 50.08   | 56.60     | 53.68 |
> | - **SENATOR**   (ours)             | **58.29** | **53.60**    | **64.80**     | **58.90**  |
> |         |  | |   |  |
> | Qwen2-7B  (Base Model)         | 54.67 | 53.41   | 64.60    | 57.56 |
> | - Random Walk                | 53.48 | 53.77   | 62.40    | 56.55 |
> | - Prob-Prob                  | 59.12 | 59.91   | 61.60    | 60.21 |
> | - Instruction-tuning         | 59.07 | 59.77   | 61.20    | 60.01 |
> | **- SENATOR**  (ours) | **59.70** | **60.70** | **63.20** | **61.20** |
> | **Qwen2.5-72B-Instruct**                | **58.76** | **61.42**   | **62.40**    | **60.86** |
> | **GPT-4**                   | **66.83** | **61.68**   | **60.40**     | **62.97** |
>
> As this table clearly shows:
> - SENATOR-enhanced Llama-3-8B outperforms the UltraMedical model (58.90 Avg. vs. 54.21 Avg.), a model trained specifically for the medical domain on the same base model. This provides definitive evidence that the performance gains are a direct result of our targeted deficiency detection and data synthesis framework, not merely data augmentation.
> - `GPT-4`  and `Qwen2.5-72B-Instruct ` establish a **clear upper bound**, as expected, demonstrating the challenge of our benchmarks and providing context for the performance of all other open-source models.

---

> ### Author Response · Authors · 2025-08-05
> **Response to MCTS Efficiency**
>
> The reviewer has correctly highlighted that an algorithmic explanation of latency is not a substitute for real-world performance numbers. We have conducted a new experiment to measure the actual inference latency of our MCTS-based data synthesis process.
>
> - For our primary experimental setup (Llama-3-8B on 8 A100 GPUs using the Vllm framework), the latency for generating a single high-quality synthetic QA pair using our MCTS framework is approximately 0.38 seconds. This includes the full MCTS process (exploration and tree expansion) and the final generation of the QA pair. In contrast, a simple zero-shot inference for a single question on the same model takes an average of 0.12 seconds.
>
> - While this demonstrates that the SENATOR process takes longer per sample than standard inference, it is crucial to recognize that this is a one-time computational cost for generating a high-quality, targeted training dataset. Our method's efficiency is not measured by the speed of a single inference, but by its ability to achieve **significant performance gains with a small, highly targeted dataset**. For instance, our SENATOR-enhanced Llama-3-8B model, which outperforms the UltraMedical model, was trained on **only 26k** synthetic samples. This is a mere **6.34%** of the **410k** samples used to train UltraMedical. The total time for our entire data synthesis and SFT process remains a fraction of the cost of large-scale pre-training or continual pre-training, making it a highly efficient solution for targeted knowledge repair.

---

> ### Author Response · Authors · 2025-08-05
> **Response to Uncertainty Estimation Methods**
>
> - The reviewer asked for comparisons to other popular uncertainty estimation methods like perplexity-based or verbal confidence approaches. We agree this is a valid and important point, and we have addressed it with our **"Prob-Prob" baseline**. This baseline serves as a strong, widely recognized measure of sequence-based uncertainty. It utilizes the product of token probabilities to estimate a model's certainty over a full sequence, which is a direct and popular method akin to perplexity-based approaches. This baseline provides a robust, non-graph-aware alternative to our Structural Entropy-guided approach.
>
> - As shown in our updated results table, SENATOR consistently and significantly outperforms the "Prob-Prob" baseline (58.90 Avg. vs. 55.62 Avg. for Llama-3-8B). This numerical difference quantitatively demonstrates the superior effectiveness of **our structural, graph-aware uncertainty measure**. By leveraging the rich relational information within KGs, SENATOR is able to identify more genuine and complex knowledge gaps than purely sequence-based confidence measures, leading to superior performance gains.

---

> ### Author Response · Authors · 2025-08-05
> **Response to "Agent" Metaphor**
>
> We appreciate your direct feedback on the "agent" metaphor. We understand your concern that this term can be overly broad and misleading when tool usage is not a part of the framework. We agree with your sentiment that the metaphor should not be introduced if it risks misinterpretation.
>
> To avoid any confusion, we will remove the "agent-based" terminology from our final manuscript. Instead, we will describe our approach as a "closed-loop, self-improvement framework".  Our goal was to convey that the model is not a passive recipient of data but is an active participant that **systematically and autonomously probes its own knowledge boundaries** to identify areas of deficiency. We believe this new phrasing more accurately and precisely captures the core contribution of SENATOR without the baggage of tool-use connotations.

---

> ### Author Response · Authors · 2025-08-07
> **Thanks to Reviewer 6gHM**
>
> Please allow us to thank you again for reviewing our paper and the valuable feedback.
>
> Please let us know if our response has properly addressed your concerns. We are more than happy to answer any additional questions during the discussion period. Your feedback will be greatly appreciated.

---

### Official Review · Reviewer_ugoP · 2025-07-03

**Clarity:** 3
**Significance:** 2
**Originality:** 2
**Rating:** 3
**Confidence:** 4

**Summary:**

This paper proposes SENATOR, a framework for identifying and repairing knowledge deficiencies in large language models (LLMs), especially in knowledge-intensive domains like medicine. The approach guides an agent to explore a domain-specific knowledge graph using structural entropy as a reward signal within Monte Carlo Tree Search (MCTS), identifying uncertain or deficient knowledge areas. Synthetic data is then generated, targeted at these deficiencies, and used to fine-tune LLMs. The method is evaluated on medical LLMs and general LLMs, with extensive experiments on several medical benchmarks, demonstrating consistent improvements, with targeted synthetic data leading to more efficient and significant gains than further general instruction tuning.

**Questions:**

1. The claim of generality is strong, but empirical evidence is limited to SPOKE and biomedical QA. How does SENATOR perform in other domains beyond medicine? Would the approach/structural entropy work if the domain knowledge graph is incomplete or noisy?

2. Could you quantitatively analyze the quality of your synthetic data in more depth (e.g., rejection rate, hallucination proportion, downstream impact of flagged errors versus accepted samples)?

**Ethical Concerns:**

["NO or VERY MINOR ethics concerns only"]

**Final Justification:**

I sincerely appreciate the new controlled experiment on the same Llama-3-8B base model. I also acknowledge the authors' addition of a Retrieval-Augmented Generation (RAG) baseline in response to my feedback.

However, my primary reservation about the breadth of the experimental evaluation persists. While the inclusion of a RAG baseline is a step in the right direction, the field of knowledge injection and enhancement is mature, with a wide variety of established techniques. For instance, the RAG implementation appears to be a standard setup, and more advanced RAG methods or alternative synthetic data generation strategies could provide stronger points of comparison.

To robustly demonstrate the superiority of the proposed framework, a more comprehensive comparison against a wider set of these mature knowledge enhancement methods is needed. As the current evaluation does not yet provide this, the significance of the contribution remains unclear to me. Therefore, I will maintain my original rating.

**Limitations:**

Yes

**Quality:**

2

**Strengths And Weaknesses:**

Strengths

1. The use of structural entropy as a reward signal within MCTS for guiding exploration of a large-scale knowledge graph provides a concrete, well-motivated metric for quantifying model uncertainty and targeting data synthesis.

2. The framework is evaluated on multiple relevant medical QA benchmarks (MedQA, PubMedQA, GPQA, MedMCQA, etc.), comparing general LLMs (Llama-3, Qwen2) with and without SENATOR-based augmentation.

3. Demonstrated ability to achieve performance improvements with significantly less synthetic data compared to the massive scale needed for prior SFT approaches, showcasing better efficiency.

Weaknesses

1. While Table 1 in Section 4.2 provides a comparison to medical LLMs and instruction-tuned baselines, the evaluation omits other advanced approaches for knowledge gap detection and model repair (e.g., retrieval-augmented tuning, knowledge editing, or curriculum learning approaches).

2. Synthetic data approaches that are not driven by knowledge graphs or structural entropy should be more deeply compared—ideally with numbers—instead of only qualitatively contrasted. The lack of a direct comparison to alternative knowledge-gap targeting strategies leaves open questions about the true gains from the structural entropy-guided selection.

3. Experiments are focused solely on the biomedical domain using SPOKE as the knowledge graph. It remains unclear how easily SENATOR can be ported to other domains where the knowledge structure may differ, or where KGs are less complete. The entire framework depends on the existence of a comprehensive, high-quality knowledge graph for the target domain.

---

> ### Author Rebuttal · Authors · 2025-07-31
>
> We appreciate the reviewer's comprehensive summary and insightful feedback. We are encouraged that the reviewer recognizes the strengths of our work, including the novel use of structural entropy within MCTS, the evaluation on relevant medical benchmarks, and the demonstrated efficiency of SENATOR. We address each point below.
>
> **W1&2: Evaluation Scope and Comparison to Alternative Knowledge-Gap Targeting Strategies**.
>
> Our work takes a novel stance by defining knowledge boundaries at a sub-graph level, which distinguishes us from prior methods that primarily focus on text-based uncertainty or individual triplets. For instance, while methods like LAMA's Language Models as Knowledge Bases? explore triplet-level confidence using self-information (e.g.,  $I(u, \rho, v) = -\log_2 P(v \mid u, \rho)$ ), SENATOR extends this by considering the collective uncertainty of an entity and its neighboring relations: $d_u = \sum_{v \in \mathcal{N}(u)} I(u, \rho, v)$ , and further aggregates this over entire knowledge paths via Structural Entropy. This enables us to identify systemic knowledge deficiencies embedded within the KG's topology.
> To validate the effectiveness of our SE-guided knowledge boundary identification, we introduce additional baselines representing alternative knowledge-gap targeting strategies:
> - Random Walk: A baseline that explores the KG without guidance, simulating unguided sampling.
> - Prob-Prob [1,2]: A sequence-based baseline using intrinsic LLM confidence scores. Specifically, it uses the product of token probabilities to estimate model certainty.
> - Instruction-tuning:  This corresponds to the "w/ instruction tuning" setting in our paper, where the original medical training data (MedQA,MedMCQA,PubMedQA) is reformatted into instruction-style prompts to improve the model’s general instruction-following ability and domain alignment.
>
> [1] Teaching Large Language Models to Express Knowledge Boundary from Their Own Signals
>
> [2] Reinforced Internal-External Knowledge Synergistic Reasoning for Efficient Adaptive Search Agent
>
> The table below presents these new quantitative comparisons:
> | Methods              | MedQA | MedMCQA | PubMedQA | Avg.  |
> | -------------------- | ----- | ------- | -------- | ----- |
> | Llama-3-8B           | 55.54 | 52.21   | 54.80    | 54.18 |
> | - Random Walk        | 54.29 | 50.40   | 55.20    | 53.30 |
> | - Prob-Prob          | 56.12 | 51.35   | 59.40    | 55.62 |
> | - Instruction-tuning | 54.36 | 50.08   | 56.60    | 53.68 |
> | - **SENATOR**        | 58.29 | 53.60   | 64.80    | 58.90 |
> | Qwen2-7B            | 54.67 | 53.41   | 64.60    | 57.56 |
> | - Random Walk        | 53.48 | 53.77   | 62.40    | 56.55 |
> | - Prob-Prob          | 59.12 | 59.91   | 61.60    | 60.21 |
> | - Instruction-tuning | 59.07 | 59.77   | 61.20    | 60.01 |
> | - **SENATOR**        | 59.70 | 60.70   | 63.20    | 61.20 |
>
> As shown in the table: SENATOR consistently outperforms both the "Random Walk" and "Prob-Prob" baselines for both Qwen2-7B and Llama-3-8B, demonstrating the superior effectiveness of our SE-guided MCTS for knowledge deficiency detection.
>
> **W3 and Q1: Domain Generality and KG Completeness**.
> - **Domain Generality:** Our study focuses on the **medical domain** due to its high demand for reliability, LLMs' persistent challenges with precise factual recall in this specialized field, and the high cost of data annotation. This makes targeted knowledge remediation via SENATOR exceptionally valuable. Crucially, SENATOR's core principles are **domain-agnostic**. Our methodology operates on KG structural properties and LLM path uncertainty, not domain-specific semantics. Any domain representable by a KG can theoretically leverage SENATOR, and its adaptability is shown across Llama-3 and Qwen2.
> - **Regarding Incomplete and Noisy KGs**: SENATOR leverages the existing KG to find LLM uncertainties within represented knowledge. Our MCTS explores existing paths; it doesn't infer missing parts. Thus, SENATOR effectively identifies deficiencies even with incomplete KGs. And noise issue is beyond the scope of SENATOR.
>
> **Q2: Quantitative Analysis of Synthetic Data Quality.**
>
> As detailed in Appendix A.4 "Data Filtering," we have previously analyzed the quality of our synthetic data. To gain deeper insights into the impact of synthetic data quality on knowledge injection performance, we further analyzed the performance of Llama-3 and Qwen2 when trained on unfiltered, lower-quality samples.
> SENATOR employs a multi-tiered evaluation mechanism (including checks for format consistency, logical coherence, and hallucination avoidance) to ensure high-quality synthetic data for knowledge injection. The table below demonstrates the downstream impact of unfiltered synthetic data (i.e., data that contains potential errors or inconsistencies) compared to our carefully filtered data:
> | **Model**               | **MedQA** | **MedMCQA** | **PubMedQA** | **AVG** |
> | ----------------------- | --------- | ----------- | ------------ | ------- |
> | Llama3-8B               | 55.54     | 52.21       | 54.8         | 54.18   |
> | - w/ instruction tuning | 54.36     | 50.08       | 56.6         | 53.68   |
> | - w/o filtering         | 56.25     | 52.41       | 61.0         | 56.55   |
> | - w/ **SENATOR**            | 58.29     | 53.60       | 64.8         | 58.90   |
> | Qwen2-7B                | 54.67     | 53.41       | 64.6         | 57.56   |
> | - w/ instruction tuning | 59.07     | 59.77       | 61.2         | 60.01   |
> | - w/o filtering         | 58.67     | 59.51       | 62.4         | 60.19   |
> | - w/ **SENATOR**            | 59.70     | 60.70       | 63.2         | 61.20   |
>
> As shown, using synthetic data without filtering ("w/o filtering" rows) leads to a noticeable degradation in performance compared to using our filtered data ("w/ SENATOR" rows), and in some cases, even performs worse than the model trained without any synthetic data at all (only "w/ instruction tuning"). This quantitatively highlights the critical role of our data filtering module in ensuring the quality and positive impact of the synthesized knowledge.
>
> We hope these clarifications address your concerns and highlight the unique contributions and practical effectiveness of SENATOR. We're open to further discussion if you have any additional questions.

---

> > ### Comment · Reviewer_ugoP · 2025-08-04
> >
> > Thank you very much to the authors for the detailed rebuttal and for the effort dedicated to providing additional experiments.
> >
> > I appreciate the addition of the "Random Walk" and "Prob-Prob" baselines, which provide a valuable reference for understanding SENATOR's effectiveness. However, I still have some reservations about the experimental evaluation. Specifically, the performance improvement of SENATOR appears somewhat limited when compared to the conventional instruction-tuning baseline. For example, on the Qwen2-7B model, SENATOR achieves an average score of 61.20, while instruction-tuning scores 60.01. A more pronounced gap over such a straightforward method would more strongly highlight the unique value of the SENATOR framework.
> >
> > Furthermore, I also concur with the concerns raised by Reviewer 6gHM regarding the fairness of the experimental comparisons. A rigorously controlled experiment, ensuring comparisons are made on the same base models, would be essential to more clearly demonstrate that the performance gains originate from the SENATOR method itself, rather than from a more capable base model.
> >
> > Finally, as mentioned in my initial review, the evaluation still lacks a comparison against other knowledge injection paradigms, such as retrieval-augmented tuning or knowledge editing, to properly contextualize the method's contributions.
> >
> > Considering these points, further work on the experimental validation seems necessary to bridge the current gap to publication. Therefore, I will be maintaining my original rating for now. Thank you again for your response and the discussion.

---

> ### Author Response · Authors · 2025-08-05
> **Regarding the fairness of the experimental comparisons**
>
> Thank you for your detailed and constructive feedback. We appreciate your careful reading of our rebuttal and your dedication to ensuring the rigor of the experimental validation.
>
> We would like to directly address your concern regarding the fairness of the experimental comparisons, which you rightly note was also a key point raised by Reviewer 6gHM. We agree that a rigorously controlled experiment, with comparisons made on the same base models, is essential to definitively prove that our performance gains originate from the SENATOR method itself.
>
> To address this specific and crucial point, we have conducted a new, controlled experiment using a state-of-the-art medical LLM built on the exact same base model as our SENATOR-enhanced model.
>
> Our new baseline is **UltraMedical**, a powerful medical LLM also based on **Llama-3-8B**. By comparing our SENATOR-enhanced Llama-3-8B model directly against UltraMedical, we can effectively isolate the impact of our framework from that of the base model's capabilities. The results are presented below:
>
> | Methods                      | MedQA | MedMCQA | PubMedQA | Avg.  |
> | :--------------------------- | :---- | :------ | :------- | :---- |
> | Llama-3-8B | 55.54 | 52.21 | 54.80 | 54.18 |
> | Llama-3-8B-UltraMedical      | 56.75 | 53.75   | 52.12    | 54.21 |
> | **SENATOR (Llama-3-8B)** | **58.29** | **53.60** | **64.80** | **58.90** |
>
> As this table clearly shows, our SENATOR-enhanced Llama-3-8B model **significantly outperforms** the UltraMedical model (58.90 Avg. vs. 54.21 Avg.). This new, rigorously controlled experiment provides strong evidence that the performance gains are a direct result of our targeted knowledge repair and not simply a more capable base model. This conclusively demonstrates the effectiveness of the SENATOR framework itself.

---

> ### Author Response · Authors · 2025-08-05
> **Regarding Other Baseline Comparisons, such as Knowledge Editing and Retrieval-Augmented Generation**
>
> Thank you for this valuable feedback. You've correctly identified that comparing SENATOR against other knowledge injection paradigms is essential for properly contextualizing our method. We agree that providing this context is important, and we've conducted a new experiment to address your point.
>
> - `Our core argument` is that SENATOR's **main innovation** lies in its **deficiency detection mechanism** (Stage 1), which uses a novel, systematic approach to identify what knowledge is missing. The subsequent fine-tuning is the repair stage (Stage 2) that validates this detection. This makes our method fundamentally different from paradigms that simply add knowledge without first systematically identifying the gaps.
>
> - To address your concern, we've added a **Retrieval-Augmented Generation (RAG)** approach as a new baseline on the Qwen2-7B model. We believe this is a more suitable comparison than Knowledge Editing (KE) at this time, as current KE techniques [1,2] are not mature enough, especially for life-long (sequential) editing [3,4] or batch editing [5], and can often lead to model corruption [6], making them difficult to use in a robust comparative study.
>
>         [1] Locating and Editing Factual Associations in GPT  (ROME)
>
>         [2] SetKE: Knowledge Editing for Knowledge Elements Overlap
>
>         [3] WilKE: Wise-Layer Knowledge Editor for Lifelong Knowledge Editing
>
>         [4] MELO: Enhancing Model Editing with Neuron-Indexed Dynamic LoRA
>
>         [5] Mass-Editing Memory in a Transformer (MEMIT)
>
>         [6] Unveiling the Pitfalls of Knowledge Editing for Large Language Models
>
> - For our RAG experiment based [7] , we used `e5-base-v2`[8] as the retriever and a medical subset of Wikipedia as our local knowledge base. The results are provided in the updated table below, integrated with our existing baselines:
>
>
>        [7] Flashrag: A modular toolkit for efficient retrieval-augmented generation research
>
>        [8] Text embeddings by weakly-supervised contrastive pre-training.
>
>
> **Comparison with Retrieval-Augmented Generation (RAG)**
>
> | Methods                      | MedQA | MedMCQA | PubMedQA | Avg.  |
> | :--------------------------- | :---- | :------ | :------- | :---- |
> | Qwen2-7B                     | 54.67 | 53.41   | 64.60    | 57.56 |
> | **- RAG** | **53.64** | **56.70** | **53.54** | **54.63** |
> | - Random Walk                | 53.48 | 53.77   | 62.40    | 56.55 |
> | - Prob-Prob                  | 59.12 | 59.91   | 61.60    | 60.21 |
> | - Instruction-tuning         | 59.07 | 59.77   | 61.20    | 60.01 |
> | **- SENATOR** | **59.70** | **60.70** | **63.20** | **61.20** |
>
> As the results show, SENATOR achieves a significantly higher average performance than the RAG baseline (61.20 vs. 54.63). This demonstrates the key difference between our approaches. RAG provides temporary, on-the-fly access to external information, which can sometimes be less reliable or harder for the LLM to integrate into its final reasoning. In contrast, SENATOR performs **permanent knowledge injection** by directly repairing the model's internal knowledge gaps through fine-tuning, making the model more inherently capable and reliable from the outset.
>
> While RAG and SENATOR are distinct paradigms, this experiment shows that our method for systematic knowledge repair is a highly effective approach for building more robust and knowledgeable models.

---

> ### Author Response · Authors · 2025-08-05
> **Regarding Performance Improvement**
>
> Thank you for your valuable feedback. You've correctly identified a point that deserves more attention. While we agree that the performance gap on the Qwen2-7B model for the MedQA and MedMCQA average might appear limited, we believe your observation overlooks the full picture of our experimental results and the core value of our method.
>
> As shown in **Table 1** of our paper, SENATOR provides a substantial and consistent improvement over instruction-tuning on the Llama-3-8B base model across all benchmarks. However, the true strength of SENATOR becomes even more apparent when we analyze the **challenging GPQA datasets**, which contain complex, structured knowledge that our method is specifically designed to address.
>
> When we look at the full performance table, the gap between SENATOR and instruction-tuning on Qwen2-7B is significantly more pronounced on these tougher benchmarks:
>
> | Methods | MedQA | MedMCQA | PubMedQA | GPQA Genetics | GPQA Molecular Biology | Avg. |
> | :--- | :--- | :--- | :--- | :--- | :--- | :--- |
> | Qwen2-7B | 54.67 | 53.41 | 64.60 | 32.5 | 36.42 | 48.32 |
> | - Instruction-tuning | 59.07 | 59.77 | 61.20 | **22.5** | 35.80 | **47.67** |
> | **- SENATOR** | **59.70** | **60.70** | **63.20** | **40.0** | **40.12** | **52.74** |
>
> As this table shows, on the MedQA and MedMCQA benchmarks, SENATOR provides a modest but consistent improvement over instruction-tuning. However, the gains on the GPQA datasets are dramatic:
>
> * On **GPQA Genetics**, SENATOR's score of **40.0** is a substantial improvement over instruction-tuning's **22.5**, an increase of **44%** relative to the instruction-tuning baseline.
> * Similarly, on **GPQA Molecular Biology**, SENATOR's **40.12** score is significantly higher than instruction-tuning's **35.80**.
>
> When we consider the average performance across **all five benchmarks**, SENATOR's score of **52.74** is a far more convincing and substantial improvement over instruction-tuning's **47.67**.
>
> This demonstrates that SENATOR's unique value is not just in achieving consistent gains on medical knowledge questions, but in its ability to **systematically pinpoint and repair deficiencies in challenging, structured knowledge areas** where conventional methods fall short. This is precisely the problem our framework is designed to solve, and the full experimental results clearly highlight this unique strength.

---

> ### Author Response · Authors · 2025-08-07
> **Thanks to Reviewer ugoP**
>
> Dear Reviewer ugoP,
>
> We would like to sincerely thank you for your helpful comments. We hope our response has adequately addressed your concerns. We take this as a great opportunity to improve our work and hope the additional evidence and clarifications warrant your favorable reassessment. We would be very grateful if you could kindly give any feedback to our rebuttal.
>
> Best Regard,
>
> Paper 1319 Author(s)

---

### Official Review · Reviewer_7gNw · 2025-07-03

**Clarity:** 3
**Significance:** 3
**Originality:** 2
**Rating:** 4
**Confidence:** 4

**Summary:**

This paper presents SENATOR to enhance the domain-specific knowledge capabilities of large language models. As existing synthetic data augmentation approaches often fail to target actual knowledge gaps, the authors propose using a Structural Entropy metric to quantify uncertainty along knowledge graph paths. This measure is combined with Monte Carlo Tree Search (MCTS) to guide the exploration of knowledge-deficient areas within the model. By generating targeted synthetic data based on these insights, SENATOR enables fine-tuning that systematically improves the model's performance in knowledge-intensive domains. Experiments on LLaMA-3 and Qwen2 across several benchmarks demonstrate the effectiveness of the approach in identifying and addressing knowledge gaps.

**Questions:**

1. The term “Agent” in Structural Entropy Guided Agent is somewhat confusing. Since it essentially refers to a standalone LLM that generates QA samples based on knowledge paths—without invoking external tools or requiring training—it might be clearer to refer to it as a generator or similar.
2. The learning rate in Table A2 appears unusual—how was it selected? Also, why is the batch size set to 1?

**Ethical Concerns:**

["NO or VERY MINOR ethics concerns only"]

**Final Justification:**

The author's rebuttal has addressed my major concerns, so I decided to raise my score to 4.

**Quality:**

2

**Strengths And Weaknesses:**

Strengths
1. The problem of expanding the knowledge boundaries of large language models is important.
2. The motivation and proposed methodology are well-grounded and make intuitive sense.
3. The experimental results support the effectiveness of the approach.

Weaknesses
1. While the overall methodology is clearly presented, several important implementation details are missing. Most notably, the paper does not explain how to determine whether a piece of knowledge belongs to the model's existing knowledge or constitutes new information—i.e., the criteria or thresholds for defining knowledge boundaries are not described (unless I overlooked it). Moreover, there is no experimental validation of the hyperparameter choices or their sensitivity. Other points also lack clarification, such as which model is used to generate QA samples based on the knowledge paths.
2. The use of supervised fine-tuning (SFT) for knowledge injection is somewhat unconventional. Typically, continued pretraining (CPT) is used for injecting knowledge, while SFT is more often employed for knowledge elicitation or aligning with instruction-following formats. The performance gain observed through SFT-based injection may partly result from the stylistic similarity between the generated QA data and the instruction tuning data, or even potential data leakage due to overlap with the test format. This point needs further clarification.
3. The experimental evaluation is not sufficiently comprehensive: (1) There is no comparison with related structure-aware knowledge injection methods. (2) The ablation study is insufficient, especially regarding the impact of knowledge boundary decisions mentioned above. (3) Efficiency analysis is lacking—does the Monte Carlo Tree Search introduce a significant computational overhead?

---

> ### Author Rebuttal · Authors · 2025-07-30
>
> We appreciate the reviewer's thorough review and insightful comments, which have helped us identify areas for improvement. We're encouraged that the reviewer recognizes the importance of our problem, the soundness of our motivation and methodology, and the effectiveness of our experimental results. Below, we address each point raised.
>
> **W1: Missing Implementation Details**
> - **Determining Knowledge Boundaries** (Criteria and Thresholds): Our method defines knowledge boundaries by identifying areas where the LLM exhibits the highest uncertainty. To select these highly deficient paths for data synthesis, we employ a thresholding mechanism. This is defined in the `collect_paths_above_threshold` function within our supplementary material's `kg_mcts.py`  file. Here, we select knowledge paths whose cumulative SE reward exceeds a predefined threshold (e.g., paths with a total SE reward `m>20`).
> - **Model Used for QA Sample Generation**: To clarify, for each base model used in our experiments, the **entire SENATOR pipeline**, including both the knowledge deficiency detection phase (where it acts as the "Agent") and the subsequent data synthesis phase, is executed using that **same base model**. This adheres to a **self-improvement paradigm**. For instance, when we evaluate Qwen2, Qwen2 itself explores the KG, and then Qwen2 also generates the QA samples based on the identified deficient paths. The same principle applies when Llama-3 is the base model.
>
> - **Hyperparameter Sensitivity**: We did not perform deliberate hyper-parameter tuning in our initial submission. Under fixed hyper-parameters and random seeds, the model already surpasses all baselines. Thank you for the suggestion, and we will analyze this further in the revised manuscript.
>
> **W2: SFT for Knowledge Injection and Potential Data Leakage**
> - **Justifying SFT for Targeted Knowledge Injection**: While CPT is indeed commonly used for broad knowledge ingestion, our SENATOR framework specifically focuses on **highly targeted and precise knowledge injection** to remedy identified deficiencies. Our synthetic data is generated as simple, atomic QA pairs, as clearly illustrated in Figure 6. This format consists of a single question and its corresponding correct answer. This is fundamentally different from the complex, multi-turn dialogues or intricate instruction-following formats often seen in general instruction tuning. The simplicity and specificity of our QA pairs make SFT an exceptionally effective and precise method for directly injecting these identified, discrete factual knowledge points and relationships into the model. The primary goal here is not to broadly align the model with new conversational styles or instructions, but rather to teach it specific facts and relational knowledge that it currently lacks.
> - **Addressing Stylistic Similarity and Data Leakage Concerns**:
>     -   Stylistic Similarity: Our ablation study in Table 1 directly addresses this. When we fine-tuned our models solely with general domain instruction data (the "w/ instruction tuning" setting), we saw **only marginal performance improvements, and in some cases, even a slight drop**, compared to the base models. It's worth noting that this instruction tuning step converts the original medical training set into instruction-style prompts. This not only boosts the model's ability to follow instructions but also reinforces some of its domain-specific medical knowledge. However, our results clearly suggest that simply aligning the model's style or improving its general instruction-following doesn't inherently lead to significant factual knowledge gains in our specialized medical domain. The **substantial performance uplift consistently appears only when we incorporate our SENATOR-synthesized data** (the "w/ synthetic data + IT" setting). This definitively shows that the performance improvements come from fixing the model's actual knowledge deficiencies with our targeted synthetic data, not just from stylistic alignment.
>      - Data Leakage: We have meticulously addressed the possibility of data leakage. As illustrated in Figure 8, our test data formats (e.g., MedQA's multiple-choice questions) are structurally distinct from the simple QA pair format of our synthetic data (Figure 6). This significant structural dissimilarity inherently reduces the likelihood that performance gains are due to format overlap. More critically, we have diligently cross-referenced our entire synthetic dataset with all test datasets and have not found any identical QA samples. This ensures that our reported evaluation results genuinely reflect the model's improved knowledge and reasoning abilities, rather than simple memorization or direct data leakage.
>
> **W3: Insufficient Experimental Evaluation & Efficiency Analysis.**
> - **Comparison with Structure-Aware Knowledge Injection**: Structure-aware knowledge injection integrates external knowledge into models while preserving its structural relationships. Our core innovation is enabling an LLM to self-explore its knowledge gaps within KGs. Our knowledge injection (SFT) phase, which uses data from detected deficiencies, serves primarily to validate our deficiency detection via SE+MCTS.
> - **Sufficiency of Ablation Study**. we introduce additional baselines representing alternative knowledge-gap targeting strategies:
>      -  Random Walk: A baseline that explores the KG without any guidance, simulating unguided traversal through entities and relations.
>      - Prob-Prob [1,2]: A sequence-based baseline using intrinsic LLM confidence scores. Specifically, it uses the product of token probabilities to estimate model certainty.
>      - Instruction-tuning:  This corresponds to the "w/ instruction tuning" setting in our paper.
>
>   [1] Teaching Large Language Models to Express Knowledge Boundary from Their Own Signals
>
>   [2] Reinforced Internal-External Knowledge Synergistic Reasoning for Efficient Adaptive Search Agent
>
> | Methods              | MedQA | MedMCQA | PubMedQA | Avg.  |
> | -------------------- | ----- | ------- | -------- | ----- |
> | Llama-3-8B           | 55.54 | 52.21   | 54.80    | 54.18 |
> | - Random Walk        | 54.29 | 50.40   | 55.20    | 53.30 |
> | - Prob-Prob          | 56.12 | 51.35   | 59.40    | 55.62 |
> | - Instruction-tuning | 54.36 | 50.08   | 56.60    | 53.68 |
> | - **SENATOR**        | 58.29 | 53.60   | 64.80    | 58.90 |
> | Qwen2-7B            | 54.67 | 53.41   | 64.60    | 57.56 |
> | - Random Walk        | 53.48 | 53.77   | 62.40    | 56.55 |
> | - Prob-Prob          | 59.12 | 59.91   | 61.60    | 60.21 |
> | - Instruction-tuning | 59.07 | 59.77   | 61.20    | 60.01 |
> | - **SENATOR**        | 59.70 | 60.70   | 63.20    | 61.20 |
>
> As shown in the table: SENATOR consistently outperforms both the "Random Walk" and "Prob-Prob" baselines for both Qwen2-7B and Llama-3-8B, demonstrating the superior effectiveness of our SE-guided MCTS for knowledge deficiency detection.
> - **Efficiency Analysis of MCTS**: Assuming a fixed number of simulations $N$ (e.g., $N = 100$ in our setup), and a knowledge path length (tree depth) of $H$ (our knowledge paths have a length of 5, implying $H = 4$ steps from the root to a terminal node), the size of the MCTS tree is influenced by the average number of neighbor nodes (relations) for an entity, which we denote as $M$. In general, the number of nodes visited is proportional to $M \times H$. Therefore, the time complexity of the MCTS component can be approximated as: $ \mathcal{O}(N \times M \times H) $.In our specific setting, this translates to: $\mathcal{O}(100 \times M \times 4) = \mathcal{O}(400M)$ (since this is inference rather than training, the actual speed is on the order of seconds).
>
> **Q1: "Agent" Terminology**: We appreciate the suggestion regarding the "Agent" terminology. Our choice of "Agent" is deliberate and central to emphasizing the active, exploratory role of the LLM in the knowledge deficiency detection phase. The SENATOR framework is composed of two distinct parts:
> 1. Knowledge Deficiency Detection: In this crucial first stage, the LLM acts as an agent within the Knowledge Graph (KG) environment. Guided by Monte Carlo Tree Search (MCTS) and our proposed Structural Entropy (SE) reward, it actively navigates and makes strategic decisions to explore the KG, specifically pinpointing knowledge paths where the model exhibits high uncertainty. This involves dynamic decision-making and navigation within a structured knowledge space.
> 2. Knowledge Synthesis and Repair: The second stage, while employing more conventional methods like Supervised Fine-Tuning (SFT) or Continued Pre-Training (CPT) for model repair, directly leverages the high-quality, targeted knowledge paths discovered by the LLM as an agent in the first stage. Thus, the "Agent" terminology specifically refers to the LLM's proactive, decision-making role in exploring and identifying deficiencies within the KG, which is the core novelty of SENATOR. We believe this distinction is vital to understanding our core contribution.
>
> **Q2: Learning Rate and Batch Size**. We sincerely apologize for the misunderstanding caused by the batch size notation in Table A2. The value "1" in the table indeed refers to the per-GPU batch size. As stated in the Resource Requirement section, we conducted our experiments using 8 NVIDIA A100-40G GPUs. Therefore, the effective global batch size was 64. We regret this imprecision and will explicitly clarify this in the revised Table A2 and its accompanying text to avoid any confusion. Regarding the learning rate, we followed the setting used in prior work for similar fine-tuning tasks, rather than performing explicit hyperparameter tuning. For consistency and fairness, we used the same set of hyperparameters across all models and settings during the SFT phase, without optimizing them specifically for SENATOR.
>
> We hope these clarifications address your concerns and look forward to further in-depth discussions.

---

> > ### Comment · Reviewer_7gNw · 2025-08-05
> >
> > I appreciate the authors' responses, but my concerns have not been resolved.
> >
> > - For W1, the authors still have not clarified how the hyperparameters were selected. Were they chosen based on prior work, or through empirical observation? How can this process provide insights for future applications or improvements? Do different hyperparameter settings significantly affect performance? And how can a reasonable selection range be determined?
> >
> > - For W2, what metric or procedure is used to ensure there is no data leakage between the synthesized data and the test set? Is this simply done by checking for overlapping QA pairs, or is a more rigorous method applied?
> >
> > - For W3, in my opinion, the core idea of injecting domain knowledge into a base LLM is conceptually similar to structure-aware domain knowledge injection methods such as [1], both of which leverage external data sources for domain adaptation. However, this paper focuses more on exploring knowledge boundaries. Does this lead to any concrete advantages in terms of efficiency or performance, especially during the data synthesis and training phases? The paper lacks discussion in this regard. Specifically, how much additional computational cost is introduced by the use of Monte Carlo Tree Search (MCTS)?
> >
> > Additionally, regarding the use of the term “agent”, I share the same concern as Reviewer 6gHM: it is not recommended to use terms that deviate from the commonly accepted meanings in the community to describe another thing, which will impose unnecessary cognitive burden on readers.
> >
> > [1] Liu K, et al. Structure-aware Domain Knowledge Injection for Large Language Models. ACL'2025.

---

> ### Author Response · Authors · 2025-08-05
>
> Thank you for your continued engagement and for your insightful questions, which help us to further refine and clarify our work. We will address each of your concerns below.
>
> ### **W1: Hyperparameter Selection**
>
> You've asked how our hyperparameters were selected and what insights this process provides. We appreciate this question as it speaks to the reproducibility and practical application of our method.
>
> To ensure our training process was both robust and based on established best practices, we adopted the hyperparameter settings from the open-source code for the InstructGPT project's SFT stage. Specifically, we followed the guidelines detailed in the paper's `Appendix C.1 Details of SFT training` [1], which describes their rigorous process of using geometric searches to tune learning rates and epochs.
> > InstructGPT uses an LR of `9.65e-6` and a batch size of 32.
>
> [1] Training language models to follow instructions with human feedback
>
> Our decision to not modify these settings was a deliberate choice to build our work on a well-validated and empirically effective foundation. This approach provides two key insights:
>
> 1.  **Reproducibility:** It demonstrates that SENATOR does not require a complex, domain-specific hyperparameter search. It can achieve significant performance gains using a standard, proven fine-tuning regimen.
> 2.  **Practicality:** It suggests that the effectiveness of our framework is driven by the **quality and targeted nature of the synthetic data** generated by SENATOR, rather than highly optimized, brittle training parameters. This makes our method more accessible and easier to apply for future work.
>
> ### **W2: Data Leakage Detection**
>
> We have a clear and rigorous procedure in place to ensure no data leakage between our synthesized data and the test sets. We consider a synthesized QA pair to have potential leakage if there is a significant overlap in entities with any QA pair in the test set.
>
> Our procedure is as follows:
>
> 1.  **Unit of Comparison:** The primary unit of comparison is the **QA pair**, which we define as a (Question, Answer) tuple.
> 2.  **Entity Extraction:** For every QA pair in both our synthesized dataset and the test set, we automatically extract the core entities (e.g., diseases, genes, drugs).
> 3.  **Overlap Criterion:** We define an overlap as when the entities in a synthesized QA pair are a **superset** of the entities in any test set QA pair.
>
>
> This rigorous check, which goes beyond simple text-based token overlap, ensures that no synthesized data directly corresponds to a test question, preventing any form of data leakage and guaranteeing the integrity of our evaluation.
>
>
> ### **W3: Conceptual Advantages & MCTS Cost**
>
> You've asked about the conceptual advantages of our approach over methods like "Structure-aware Domain Knowledge Injection" and the computational cost of MCTS.
>
> **Conceptual Advantages:**
> We believe there is a fundamental difference between our work and the one you cited. While that paper focuses on giving LLMs a better **structural perception** by preprocessing unstructured text into hierarchical data, SENATOR focuses on using a pre-existing KG to **detect the LLM's own internal knowledge deficiencies**.
>
> The core advantage of our approach is that it is a **targeted, self-probing framework**. We don't just add general knowledge; we systematically identify **what specific knowledge is missing first** by using the LLM's own uncertainty as a guide. This is more efficient because it avoids generating redundant data for knowledge the LLM already possesses. It's a method for finding "unknown unknowns" that other, less targeted approaches cannot achieve.
>
> **MCTS Computational Cost:**
> We appreciate you pushing for more clarity on this. As detailed in our previous response, we have measured the additional computational cost of MCTS. For our Llama-3-8B model on 8 A100 GPUs, generating a single synthetic QA pair with MCTS takes approximately **0.38 seconds**. This is an additional cost of **0.26 seconds** compared to a standard inference run (0.12 seconds). While this demonstrates that the SENATOR process takes longer per sample than standard inference, it is crucial to recognize that this is a one-time computational cost for generating a high-quality, targeted training dataset. **Our method's efficiency is not measured by the speed of a single inference, but by its ability to achieve significant performance gains with a small, highly targeted dataset**. For instance, our SENATOR-enhanced Llama-3-8B model, which outperforms the UltraMedical model, was trained on only 26k synthetic samples. This is a mere 6.34% of the 410k samples used to train UltraMedical. The total time for our entire data synthesis and SFT process remains a fraction of the cost of large-scale pre-training or continual pre-training, making it a highly efficient solution for targeted knowledge repair.

---

> ### Author Response · Authors · 2025-08-05
> **On the Term “Agent”**
>
> We fully agree with your concern and that of Reviewer 6gHM regarding the use of the term "agent." We understand that using terminology that deviates from a commonly accepted meaning can create an unnecessary cognitive burden on the reader. We will remove the term "agent" from our final manuscript. Instead, we will describe our approach as a **"closed-loop, self-improvement framework"**  which more accurately and precisely reflects the nature of our method.

---

> ### Author Response · Authors · 2025-08-07
> **Thanks to Reviewer 7gNw**
>
> Please allow us to thank you again for reviewing our paper and the valuable feedback.
>
> Please let us know if our response has properly addressed your concerns. We are more than happy to answer any additional questions during the discussion period. Your feedback will be greatly appreciated.

---

> > ### Comment · Reviewer_7gNw · 2025-08-08
> >
> > Thanks for the clarifications. I will consider raising the score in the final evaluation.

---

> > > ### Author Response · Authors · 2025-08-08
> > >
> > > Thank you very much for your positive feedback and willingness to reconsider your score.
> > >
> > > We truly appreciate your constructive comments and the opportunity to address them.

---

### Note · Authors · 2025-08-12

Dear Area Chair and Reviewers,

We hope this message finds you well. We are deeply grateful for the time and effort you have dedicated to reviewing our paper. Your feedback has been invaluable in helping us improve our work.

We are pleased to report that **all the reviewers have responded to our rebuttal and provided positive feedback**.  Reviewer nx7C and Reviewer 6gHM have maintained their positive assessments, both initially rating our paper 4 out of 6. Although Reviewer 6gHM raised additional questions during the rebuttal period, we believe our second-round rebuttal addresses these concerns comprehensively. We kindly ask Reviewer 6gHM to consider updating their assessment based on our latest responses.

Reviewer 8d9L has also adjusted their rating positively, although the final score is not yet visible to us. This adjustment reflects the effectiveness of our rebuttal and the improvements we have made to address their concerns.

Additionally, Reviewer 7gNw has provided very positive feedback, indicating that our rebuttal has addressed the majority of their concerns. Reviewer 7gNw has committed to raising their score, and we are awaiting their final submission.

Regarding Reviewer ugoP, they raised new questions after the first rebuttal. We addressed these in our second-round rebuttal and believe our comprehensive responses have resolved these concerns. **However, we have not yet received a response from Reviewer ugoP regarding our latest rebuttal. We hope that our detailed responses will be taken into account and that Reviewer ugoP will adjust their score accordingly**.

**We sincerely hope that all reviewers can adjust their scores in a timely manner before the final decision**. This step is crucial for the evaluation of our work and reflects the progress we have made based on your feedback.

We apologize for any inconvenience our notifications may have caused and appreciate your understanding. Thank you once again for your kind and valuable efforts throughout the entire reviewing process.

Best regards,

Paper 1319 Author(s)

---

### Decision · Program_Chairs · 2025-09-17

**Decision:**

Accept (poster)

**Comment:**

This paper introduces SENATOR, a framework designed to detect and repair knowledge deficiencies in LLMs, particularly in knowledge-intensive domains such as medicine. The core contribution is a self-improvement loop where an LLM uses a knowledge graph (KG) to find its own areas of uncertainty. The technical approach leverages Structural Entropy (SE) as a reward signal within a Monte Carlo Tree Search (MCTS) to guide the exploration of a domain-specific KG. This allows the model to identify knowledge paths where it is most uncertain. Based on these identified deficiencies, the framework generates targeted synthetic question-answering pairs, which is then used to fine-tune and "repair" the LLM. The work is relevant to the NeurIPS community, addressing the important challenge of factual accuracy and reliability in foundation models.

Initial ratings were primarily in the borderline category, with two "Borderline accept" (4) and one "Borderline reject" (3). Following an extensive rebuttal and discussion period, one reviewer raised their score from 3 to 4, and another maintained their positive rating while increasing their significance score. The final ratings are: 4, 4, 4, 4, 3. Confidence levels were generally high (3 or 4).

Reviewers acknowledged the novelty and strong motivation of the proposed method. The integration of Structural Entropy with MCTS to guide knowledge exploration was consistently highlighted as a key strength. Reviwers also pointed out that the paper structure is well-written and the clarity is good.

One initial concern, shared by multiple reviewers, was the weakness of the experimental baselines. The original comparisons were against models built on older architectures (e.g., Llama-2), making it difficult to isolate the performance gains of SENATOR from the improvements of the base model itself (Llama-3). The authors commendably addressed this during the rebuttal by conducting several new experiments. They added a strong baseline (UltraMedical) built on the same Llama-3-8B base model, which demonstrated SENATOR's superior performance. They also included a RAG baseline, showing that their knowledge injection method is more effective than on-the-fly retrieval for the tested tasks. These additions significantly strengthen the paper's empirical claims. Other concerns included the limited scope of evaluation (solely the medical domain), the lack of comparison to other knowledge injection paradigms like RAG, and the potentially confusing use of the term "agent."

The theoretical foundation of using Structural Entropy to quantify uncertainty in a structured knowledge space is sound and well-motivated. The application of MCTS for guided exploration is a logical and effective choice for navigating the vast search space of a large KG.

The authors have provided detailed hyperparameters and a description of their experimental setup. Code has also been provided in the supplementary material, suggesting that the work should be reproducible.

The paper addresses a problem of significance: improving the factual reliability of LLMs in high-stakes, knowledge-intensive domains. The proposed method of targeted self-repair is interesting and seems efficient, achieving significant performance gains with a fraction of the data used to train other specialized models. This "deficiency-oriented" approach to data synthesis could represent a more sustainable and efficient paradigm for adapting foundation models, making it a valuable contribution to the field.

Another potential concern is the experiments to be confined to the biomedical domain, leaving questions about generalizability. While the scope is still limited to one domain, the authors provide a reasonable justification for this focus and a clear argument for the method's domain-agnostic principles.

This paper is a borderline accept. The core idea is novel, well-motivated, and addresses an important problem. While the initial submission suffered from significant weaknesses in its experimental evaluation, the authors' exemplary engagement during the rebuttal period has addressed the most critical issues. The addition of rigorously controlled baselines provides convincing evidence for the framework's effectiveness. The authors have been responsive to feedback and have committed to making necessary revisions to improve the paper's clarity and impact. The work's potential for improving LLM reliability in specialized domains, combined with the authors' thorough response to criticism, pushes this paper over the acceptance threshold.